# Conformal Arbitrage: Risk-Controlled Balancing of Competing Objectives in Language Models

**William Overman**
Stanford
Graduate School of Business
wpo@stanford.edu

**Mohsen Bayati**
Stanford
Graduate School of Business
bayati@stanford.edu

## Abstract

Modern language-model deployments must often balance *competing objectives*—for example, helpfulness versus harmlessness, cost versus accuracy, and reward versus safety. We introduce **Conformal Arbitrage**, a post-hoc framework that learns a data-driven threshold to mediate between a *Primary* model optimized for a primary objective and a more conservative *Guardian*—which could be another model or a human domain expert—aligned with a guardrail objective. The threshold is calibrated with conformal risk control, yielding finite-sample, distribution-free guarantees that the long-run frequency of undesirable events (such as factual errors or safety violations) does not exceed a user-specified quota. Because Conformal Arbitrage operates wholly at the API level—without requiring access to model logits or updating model weights—it complements weight-based alignment techniques and integrates seamlessly with existing cost-aware cascades. Empirically, Conformal Arbitrage traces an efficient frontier, allowing users to define an acceptable performance level for one objective while maximizing utility in another. We observe that our method outperforms (in terms of accuracy on multiple-choice style questions) cost-matched random routing between models. These properties make Conformal Arbitrage a practical, theoretically grounded tool for trustworthy and economical deployment of large language models across a broad range of potentially competing objectives.

## 1 Introduction

Large language models (LLMs) excel at reasoning, coding, and open-domain question answering, yet real-world deployments frequently need to navigate tensions between potentially competing objectives such as *helpfulness* and *harmlessness* or *cost* and *accuracy*.

Current practices mostly tackle the tension between helpfulness and harmlessness by *modifying the model itself*: reinforcement learning from human feedback (RLHF) (Christiano et al., 2017; Ouyang et al., 2022), direct–preference optimisation (DPO) (Rafailov et al., 2023), Constitutional AI (Bai et al., 2022b), or multi-objective fine-tuning (Zhou et al., 2023; Wang et al., 2024) each produce a *single* operating point along the Pareto frontier. While powerful, these methods demand expensive data collection, GPU-intensive retraining, and — for API-only models — are often not applicable.

For the cost versus accuracy tradeoff, there has been significant work on cascades: a cheap model handles easy queries and defers the rest to a stronger fallback (Chen et al., 2023; Aggarwal et al., 2025; Zellinger et al., 2025). Recently, Jung et al. (2025) introduced Cascaded Selective Evaluation (CSE), calibrating per-model confidence estimators via fixed-sequence multiple testing to obtain rigorous guarantees on alignment to human pairwise preferences. However, these approaches are tailored for controlling a binary disagreement risk, while a user may be interested in controlling arbitrary guardrail metrics at deployment time.

39th Conference on Neural Information Processing Systems (NeurIPS 2025).

We introduce **Conformal Arbitrage (CA)**, a lightweight router that sits *outside* the language models. The term "arbitrage" captures how our approach exploits the performance gap between specialized models to achieve superior outcomes than naive selection between models. Given **(i)** a *Primary* model optimized for the primary objective and **(ii)** a more conservative *Guardian* model or a human domain expert, aligned with a guardrail objective, CA offers a principled alternative to randomized routing between models. Instead of merely alternating between models with some probability, CA learns a single scalar threshold on how strongly the Primary model favors its top choice over alternatives (a notion we formally define as "score gap" later in the paper). This threshold determines when the Primary model's confidence is sufficient to act upon its prediction versus when to defer to the Guardian, creating a principled decision boundary that optimizes the trade-off between objectives.

The threshold is calibrated using *conformal risk control* (CRC) (Angelopoulos et al., 2024), yielding *finite-sample, distribution-free guarantees* that the long-run frequency (or magnitude) of undesirable events never exceeds a user-specified budget $\alpha$. This enables precise control over trade-offs—users can explicitly specify how much they are willing to compromise on one objective to gain on the other. Because CA touches *no model weights*, it complements weight-based alignment and applies to closed, black-box APIs, making it a remarkably lightweight approach to achieving Pareto improvements over simple model selection strategies.

Our experiments study (i) the cost–accuracy trade-off on TruthfulQA and MMLU, and (ii) the helpfulness–harmlessness trade-off on PKU-SafeRLHF. All three benchmarks are multiple-choice settings in which the model is prompted to select from a fixed set of options. This regime is a natural fit for Conformal Arbitrage and in line with related literature such as Jung et al. (2025), which operates over binary choices. Thus we focus on the multiple-choice setting, but emphasize that the CA framework is not limited to such domains: the algorithm and theory carries over to free-text generation (see Appendix E) and broader decision-making tasks. Across all settings we evaluate, CA traces an efficient frontier that consistently dominates random or cost-matched routing baselines, while preserving finite-sample, distribution-free guarantees via CRC.

Conformal Arbitrage transforms an immutable, potentially unpredictable LLM (or a family of LLMs) into a controllable system whose risk–utility position can be *dialed after deployment*. In our experiments, we demonstrate this capability using state-of-the-art LLMs from the GPT-4.1 series, OpenAI (2025), showing how our method enables fine-grained control over various tradeoffs without modifying the underlying models. By requiring only a few hundred logged examples for calibration, CA offers a pragmatic path toward trustworthy, cost-efficient and customizable language-model services that can be adjusted to meet evolving requirements long after initial deployment.

## 2 Related work

Real–world deployments must strike a pragmatic balance between *helpfulness*—supplying users with accurate and detailed information—and *harmlessness*—avoiding policy-breaking or dangerous content. Early alignment work framed the problem as a single–objective optimization: RLHF (Christiano et al., 2017; Ouyang et al., 2022) and its variant DPO (Rafailov et al., 2023) collapse nuanced feedback into a *single* reward model and therefore deliver one operating point on the Pareto frontier. Subsequent methods introduced explicit two–factor training: RLHF on mixed helpful–harmless datasets (Bai et al., 2022a), Constitutional AI's self-revision loop (Bai et al., 2022b), and Bi-Factorial Preference Optimisation (BFPO) (Zhang et al., 2025) that casts the bi-objective RLHF loss as a direct supervised criterion. Safe-RLHF (Dai et al., 2023) separates a reward and a cost head and enforces constraints by Lagrangian relaxation, while Circuit Breakers intervene at generation time to halt policy-violating continuations (Zou et al., 2024).

The PKU-SafeRLHF benchmark (Ji et al., 2023) was specifically introduced to quantify this helpfulness-harmlessness trade-off, providing dual annotations that enable researchers to measure progress on both dimensions simultaneously. Anthropic's Constitutional AI (Bai et al., 2022b) further explores alignment by embedding principles directly into model training. More recently, the MoGU framework (Du et al., 2024) dynamically routes between model variants optimized separately for usability and safety. Empirically, while these approaches curb unsafe completions, they still lock the model into one fixed balance point between helpfulness and harmlessness.

Beyond helpfulness–harmlessness many other objectives— accuracy, cost, latency, fairness, demographic parity, domain–specific risk, etc.—can be in conflict. Many recent works have proposed

weight–based strategies to navigate the resulting frontiers between such competing objectives. Rewarded soups linearly interpolates checkpoints fine-tuned on distinct rewards to trace that surface (Ramé et al., 2023), Directional Preference Alignment adds multiple reward heads for steerable inference (Wang et al., 2024), MaxMin-RLHF learns a mixture of reward models to protect minority preferences (Chakraborty et al., 2024), and MO-DPO converts several preference signals into a closed-form multi-objective loss (Zhou et al., 2023). These approaches nevertheless share two limitations: **(i)** they require access to model weights and retraining, and **(ii)** they provide no theoretical guarantees that the inherent guardrail metrics driving the trade-off (e.g., safety, accuracy, or cost) will stay within a user-specified budget.

In contrast, our method of Conformal Arbitrage is weight-agnostic and sits *outside* the LLM. By calibrating a single threshold with conformal risk control (Angelopoulos et al., 2024), it transforms any pair of black-box models, one of which can be a human, into a *continuum* of operating points with *provable* finite-sample bounds on the chosen guardrail metric (e.g. harmlessness).

Conformal Arbitrage is thus closely tied to routing and cascade approaches that tackle cost–accuracy trade-offs (Chen et al., 2023; Yue et al., 2024; Ong et al., 2024; Aggarwal et al., 2025; Zellinger et al., 2025; Varangot-Reille et al., 2025), but can be used to tackle any potential pair of objectives that may be in tension, thus abstractly covering cost–accuracy cascades as a special case.

However, unlike these previous approaches we make no particular optimizations for any specific trade-off, including cost and accuracy, and we do not claim to out-perform such cascade systems on metrics for which they are explicitly optimized. Furthermore, compared to most routing approaches that rely on complex learned functions to distribute queries between models (Varangot-Reille et al., 2025), Conformal Arbitrage employs a principled, theoretically-grounded method using a single calibrated scalar threshold.

Scalable-oversight research explores how weaker agents or humans can be organized into critique hierarchies that amplify limited supervision. Amplification and Debate delegate verification to inexpensive judges and, under certain complexity assumptions, achieve provable "weak-to-strong" guarantees (Christiano et al., 2018; Irving et al., 2018; Burns and et al., 2023). Process supervision instead labels intermediate reasoning steps so that mistakes are caught early (Lightman et al., 2023). Self-reflection frameworks ask a model to generate critiques (and often revisions) of its own outputs (Madaan et al., 2023; Yang et al., 2024; Tang et al., 2024). Post-hoc risk control strategies in model deployment have also gained attention, particularly through moderation and oversight models deployed by industry leaders (OpenAI, 2023). Conformal Arbitrage complements these lines by offering a statistically-sound escalation rule. It lets a Primary model act autonomously as much as possible while respecting some risk budget, and otherwise it forwards a potentially much smaller slate of potential actions or outputs to a human or Guardian model. Finite-sample bounds from conformal risk control make the Guardian's load—and the residual risk—explicitly budgeted, providing a lightweight, post-hoc path to scalable oversight without touching model weights.

The underlying selective routing approach of our work resonates with classical selective prediction and reject-option frameworks initially formalized by Chow (1970) and later refined in modern selective classification research (Geifman and El-Yaniv, 2019).

Conformal prediction (CP) and its generalization, conformal risk control (CRC) (Vovk et al., 2005; Bates et al., 2021; Angelopoulos et al., 2024), provide distribution-free, finite-sample guarantees that make them generally attractive post-hoc alignment tools for high-stakes LLM deployments. For instance, Chen et al. (2025) align language models with human risk judgments by controlling tail risks such as toxicity, while Su et al. (2024) demonstrate conformal prediction applied effectively to black-box LLM APIs without internal access. Additionally, conformal risk control has been leveraged in deployment scenarios such as action deferral, illustrated by the KnowNo framework (Ren et al., 2023), which uses conformal uncertainty quantification to trigger human oversight.

Conformal prediction and conformal risk control have been used to filter low-confidence QA answers (Kumar et al., 2023), retain only entailment-supported sub-claims (Mohri and Hashimoto, 2024), and bound hallucination rates via abstention (Abbasi-Yadkori et al., 2024). Beyond marginal guarantees, conditional and adaptive CRC tighten coverage on hard prompts (Cherian et al., 2024), and sampling-based set prediction extends CP to free-text generation (Quach et al., 2024). Framing alignment as property testing, Overman et al. (2024) calibrate outputs to satisfy safety or fairness constraints

without retraining. Building on this lineage, we adapt CRC to learn a risk-calibrated switch between a Primary model and a Guardian model without retraining either model.

Conformal Arbitrage is most closely related to *Cascaded Selective Evaluation* (CSE) of Jung et al. (2025). CSE equips each judge with a confidence score, calibrates a per-judge threshold, and escalates through a cascade until some judge is confident, thereby controlling the Bernoulli risk that a machine-preferred answer disagrees with human majority. Conformal Arbitrage addresses more general tradeoffs: it controls *any* bounded guardrail loss (safety, accuracy, cost, latency, etc.) and can filter a large action space to a smaller candidate set that a Guardian or human refines, rather than abstaining on the whole instance. CSE's *Simulated Annotators* requires $K$-shot prompting (for $K$ examples of preference annotations) the model $N$ different times (for $N$ human annotators) in order to obtain an ensemble prediction *and* access to predictive probabilities extracted from the model's `logprobs`, so every judge call is multiplied many-fold and is limited to APIs that expose token-level logits. Conformal Arbitrage, by contrast, needs at most *one* call to the Primary and (when routed) *one* to the Guardian, treats the returned scores as opaque, requiring no access to logits or probabilities, and thus works with strictly black-box APIs.

# 3 Preliminaries

Conformal Arbitrage uses *conformal risk control* (CRC) to supply finite-sample, distribution-free guarantees on the guardrail metric while treating the underlying language models as black boxes. CRC extends the framework of *conformal prediction* (CP) (Vovk et al., 2005; Bates et al., 2021) from binary error control to control of *arbitrary bounded risks*. We briefly summarize both ideas.

**Conformal prediction**    Let $\mathcal{X}$ and $\mathcal{Y}$ be the input and output spaces, equipped with a joint probability distribution, and draw an exchangeable sample $(X_i, Y_i)_{i=1}^{n+1} \sim P$ where the first $n$ sample are used for calibration, and $(X_{n+1}, Y_{n+1})$ is used for testing. Given any predictor $f : \mathcal{X} \to \mathcal{Y}$ and score $s_f(x, y)$ (e.g. $|y - f(x)|$), let $q_{1-\alpha}$ be the $(1 - \alpha)$ empirical quantile of $\{s_f(X_i, Y_i)\}_{i=1}^n$. The conformal set is defined by $C(x) = \{ y \in \mathcal{Y} \colon s_f(x, y) \leq q_{1-\alpha} \}$, and enjoys the finite-sample guarantee $\Pr\{ Y_{n+1} \notin C(X_{n+1}) \} \leq \alpha$. Thus any black-box predictor attains $(1 - \alpha)$ coverage without distributional assumptions (Vovk et al., 2005; Bates et al., 2021).

**Conformal risk control**    Many real-world objectives are not binary mistakes but expectations of a task-specific loss—for example, safety-violation rate, factual errors, mean latency, or excess dollar cost. Conformal risk control (Angelopoulos et al., 2024) handles such objectives by introducing a *bounded, non-increasing* loss curve $L_i(\lambda) \in [0, B]$, where $B$ is an upper bound on the loss, for each calibration point, indexed by a tunable threshold $\lambda \in \Lambda \subset \mathbb{R}$. Defining the empirical risk $\hat{R}_n(\lambda) = \frac{1}{n} \sum_{i=1}^n L_i(\lambda)$, CRC selects

$$\hat{\lambda} \;=\; \inf\Big\{ \lambda \in \Lambda : \frac{n}{n+1}\, \hat{R}_n(\lambda) + \frac{B}{n+1} \leq \alpha \Big\}, \tag{1}$$

and proves the *finite-sample* guarantee for $\mathbb{E}\big[L_{n+1}(\hat{\lambda})\big] \leq \alpha$, again under assumption of exchangeability between the calibration data and test point. Choosing $L_i(\lambda) = \mathbb{I}\{Y_i \notin C_\lambda(X_i)\}$ recovers classical CP; alternative losses yield risk bounds tailored to deployment needs.

# 4 Methodology: conformal arbitrage

We aim to invoke a *Primary* model as often as possible (e.g. a helpfulness-maximizing or low-cost model) while ensuring, with high confidence, that a critical requirement (e.g. harmlessness, accuracy) is satisfied by routing calls to a *Guardian* model (or human) as needed. The linkage between the two models is formalized through conformal risk control (Angelopoulos et al., 2024).

## 4.1 Setting

Let $\{x_i\}_{i \geq 1}$ be an exchangeable sequence of $\mathcal{X}$-valued random variables that we refer to as *contexts*. Each context $x$ admits a finite, non-empty *action* set $A(x_i) = \mathcal{A}_i \subseteq \mathcal{A}$, where $|A(x_i)| < \infty$. Additionally, we assume the existence of two functions $L : \mathcal{X} \times \mathcal{P}(\mathcal{A}) \to \mathbb{R}$ and $U : \mathcal{X} \times \mathcal{P}(\mathcal{A}) \to \mathbb{R}$,

measuring, over subsets of the potential actions, loss for the guardrail metric and utility for the primary metric, respectively. We assume both of these functions satisfy the property that for $\mathcal{A}_1 \subseteq \mathcal{A}_2$ we have $L(x, \mathcal{A}_1) \geq L(x, \mathcal{A}_2)$ and $U(x, \mathcal{A}_1) \geq U(x, \mathcal{A}_2)$.

We assume access to two fixed, pre-trained models: $p, g : \mathcal{X} \times \mathcal{A} \to \mathbb{R}$, where $p$ is the **Primary** model (reward-seeking or cheap/low-accuracy) and $g$ is the **Guardian** model (safety-focused or costly/high-accuracy). Despite this simple interface, each model may internally implement arbitrarily complex computations—any architecture that outputs a score for each $(x, a)$ pair is admissible.

Although we write $p(x, a)$ and $g(x, a)$ as deterministic, each model call may depend on internal randomness $\zeta_P, \zeta_G$, producing scores $\tilde{p}(x, a, \zeta_P)$ and $\tilde{g}(x, a, \zeta_G)$. Such tuples $(x, \tilde{p}, \tilde{g})$ remain exchangeable across samples, so the finite-sample guarantees of conformal risk control are unaffected.

## 4.2 Calibration via conformal risk control

To calibrate our Conformal Arbitrage policy, we use conformal risk control (CRC) to calibrate a relaxation parameter $\hat{\lambda}$ that satisfies a user-defined risk budget $\alpha \in (0, 1)$, controlling how much we can trust the Primary model before deferring to the Guardian.

We begin with an exchangeable calibration set of $n$ samples:

$$\mathcal{D}^{(n)} = \left\{ (x_i, P_i, G_i) \right\}_{i=1}^{n}, \quad P_i = \{p(x_i, a)\}_{a \in \mathcal{A}_i}, \quad G_i = \{g(x_i, a)\}_{a \in \mathcal{A}_i}.$$

Each sample consists of a context $x_i$ and the scores assigned by both the Primary model and the Guardian model across the available action set $\mathcal{A}_i = A(x_i)$.

For any $\lambda \geq 0$, we define the $\lambda$-*relaxed candidate set*:

$$C_\lambda(x) = \left\{ a \in A(x) : p(x, a) \geq \max_{a' \in A(x)} p(x, a') - \lambda \right\}.$$

This set includes all actions whose Primary scores are within $\lambda$ of the top score. In particular, larger values of $\lambda$ increase the size of this set. Since all of the subsets $\mathcal{A}' \subseteq A(x)$ that we will consider will be of this form, $C_\lambda(x)$, for some $\lambda$, we adopt the notation $L_i(\lambda) = L(x_i, C_\lambda(x_i))$ and $U_i(\lambda) = U(x_i, C_\lambda(x_i))$

We then define a loss function on each calibration sample, measuring the *residual risk* that the Guardian model would assign to the best action in $C_\lambda(x_i)$:

$$L_i(\lambda) = \max_{a \in A(x_i)} g(x_i, a) - \max_{a \in C_\lambda(x_i)} g(x_i, a). \tag{2}$$

Intuitively, this loss captures how unsafe the most promising action (as judged by the Guardian) is among the candidates the Primary model would consider acceptable under $\lambda$.

To summarize overall risk, we compute the empirical average:

$$\hat{R}_n(\lambda) = \frac{1}{n} \sum_{i=1}^{n} L_i(\lambda),$$

and select the smallest $\lambda$ that satisfies the CRC inequality:

$$\hat{\lambda} = \inf\left\{ \lambda \geq 0 : \frac{n}{n+1} \hat{R}_n(\lambda) + \frac{1}{n+1} \leq \alpha \right\}. \tag{3}$$

**Definition 1** (Relaxation Parameter). *The relaxation parameter $\hat{\lambda}$ is defined as the minimal value of $\lambda$ that satisfies the conformal risk control inequality in Equation 3.*

This relaxation parameter controls the permissiveness of the candidate action set while ensuring that the expected residual risk on a new context remains bounded by $\alpha$. The guarantee holds exactly at finite sample size and requires no assumptions on score calibration or context distribution.

## 4.3 Conformal arbitrage algorithm

We now describe the deployment-time decision procedure for selecting actions using the calibrated relaxation parameter $\hat{\lambda}$ obtained in Section 4.2. At each test instance, the algorithm first consults the

---

**Algorithm 1** Conformal Arbitrage

---

**Require:** Context $x$, relaxation parameter $\hat{\lambda}$, Primary model $p$, Guardian model $g$
 1: Compute $p(x, a)$ for all $a \in \mathcal{A}(x)$
 2: Let $C_\lambda(x) = \left\{ a \in A(x) : p(x, a) \geq \max_{a'} p(x, a') - \hat{\lambda} \right\}$
 3: **if** $|C_\lambda(x)| = 1$ **then**
 4:     **return** the unique element of $C_\lambda(x)$
 5: **else**
 6:     Compute $g(x, a)$ for all $a \in C_\lambda(x)$
 7:     **return** $a^\star = \arg \max_{a \in C_\lambda(x)} G(a)$
 8: **end if**

---

Primary model to form a $\hat{\lambda}$-relaxed candidate set. If the top action is sufficiently dominant (i.e., the set is a singleton), it is selected; otherwise, the Guardian model selects from the $\lambda$-relaxed set. The procedure is outlined in Algorithm 1.

Although we present the algorithm assuming a predefined action set $\mathcal{A}(x)$, the same formulation applies directly to free-text generation, where the potential action space (all strings up to some maximum length $L$) is combinatorially large but still finite. In that case, the Primary's fixed generation policy induces a finite slate

$$\mathcal{S}(x) = \{a_1, \ldots, a_K\} \subseteq \mathcal{Y}_{\leq L},$$

and the Conformal Arbitrage procedure operates identically with Primary-model scores defined on $\mathcal{S}(x)$ and set to $-\infty$ for all actions in $\mathcal{A}(x) \setminus \mathcal{S}(x)$. This instantiation is discussed in further detail in Appendix E.

The guarantee that Algorithm 1 enforces an upper bound on the expected guardrail loss,

$$\mathbb{E}[L(x, C_{\hat{\lambda}}(x))] \leq \alpha,$$

is a direct corollary of Theorem 1 in Angelopoulos et al. (2024), which establishes finite-sample, distribution-free validity of conformal risk control under exchangeability. Intuitively, this ensures that the long-run expected violation of the guardrail metric—whether safety, factuality, or any bounded risk measure—remains below the user-specified budget $\alpha$ on unseen test contexts.

**Corollary 1** (Guardrail control under Conformal Arbitrage). *Let $(x_i, P_i, G_i)_{i=1}^{n+1}$ be an exchangeable sequence and let $\hat{\lambda}$ be the relaxation parameter obtained by conformal risk control as in (3). Then Algorithm 1 satisfies*

$$\mathbb{E}\big[L(x_{n+1}, C_{\hat{\lambda}}(x_{n+1}))\big] \leq \alpha,$$

*where the expectation is taken over the calibration and test samples. That is, Conformal Arbitrage inherits the same finite-sample, distribution-free guardrail guarantee from Theorem 1 of Angelopoulos et al. (2024).*

### 4.4 Optimality amongst score-gap routers

To address utility as measured by the primary metric we define the following class of policies, "Score-gap routers," in Definition 2. Additionally, for this theoretical result, we will require a stronger assumption of i.i.d. on the calibration data and test point.

**Definition 2** (Score-gap router). *Fix a **Primary** score function $p : \mathcal{X} \times \mathcal{A} \to \mathbb{R}$ and a non–negative threshold $\lambda \geq 0$. For each context $x$ let*

$$a^\star(x) = \arg\max_{a \in A(x)} p(x, a), \quad \Delta(x) = p\big(x, a^\star(x)\big) - \max_{b \in A(x) \setminus \{a^\star(x)\}} p(x, b),$$

*with the convention $\Delta(x) = +\infty$ if $|A(x)| = 1$. The score-gap router with threshold $\lambda$, $\mathcal{R}_\lambda : \mathcal{X} \to \mathcal{A} \cup \{\text{DEFER}\}$ acts as*

$$\mathcal{R}_\lambda(x) = \begin{cases} a^\star(x), & \textit{if } \Delta(x) \geq \lambda, \\ \text{DEFER}, & \textit{otherwise}, \end{cases}$$

*where DEFER means "forward this instance to the **Guardian** model."*

Given the Primary model's confidence scores $p(x, a)$, it chooses the top-scoring action whenever its margin over every alternative exceeds a non-negative threshold $\lambda$, and **defers** to the Guardian otherwise. This rule mirrors Chow's Bayes-optimal *reject-option* classifier (Chow, 1970): rather than rejecting an uncertain instance we escalate it to a more conservative model.

Theorem 1 establishes that *no other Score-gap router of the Primary scores alone* can deliver strictly higher expected primary utility while still obeying the same guardrail risk budget $\alpha$, up to a vanishing $O(n^{-1})$ term. We let our Primary metric be measured by $U(\lambda) = \mathbb{E}[U_i(\lambda)]$, which we assume to be non-increasing and $K$-Lipschitz. This is natural as raising $\lambda$ can only shrink the set of contexts on which we choose the Primary model's output. The proof of Theorem 1 is provided in Appendix A.1.

**Theorem 1** (Utility–optimality of Conformal Arbitrage). *Fix a compact interval $\Lambda = [0, \lambda_{\max}]$. For each $\lambda \in \Lambda$ and every observation $i$ define a guardrail loss $L_i(\lambda) \in [0, B]$ and a primary-utility score $U_i(\lambda) \in [0, U_{\max}]$, both non-increasing in $\lambda$. Write*

$$R(\lambda) = \mathbb{E}[L_i(\lambda)], \qquad U(\lambda) = \mathbb{E}[U_i(\lambda)].$$

*Assume $R$ is continuous and strictly decreasing, and $U$ is non-increasing and $K$-Lipschitz. For a desired risk budget $\alpha \in (0, B)$ let $\lambda_\star = \inf\{\lambda \in \Lambda : R(\lambda) \leq \alpha\}$. Given an i.i.d. calibration sample $\mathcal{D}^{(n)}$ of size $n$, set*

$$\widehat{R}_n(\lambda) = \frac{1}{n} \sum_{i=1}^{n} L_i(\lambda), \qquad \hat{\lambda} = \inf\Big\{\lambda \in \Lambda : \tfrac{n}{n+1} \widehat{R}_n(\lambda) + \tfrac{B}{n+1} \leq \alpha\Big\}.$$

*Then, with expectation taken over the calibration sample*

$$\mathbb{E}\big[U(\lambda_\star) - U(\hat{\lambda})\big] = O(n^{-1}),$$
$$\mathbb{E}\Big[\sup_{\substack{\tilde{\lambda} \in \Lambda \\ R(\tilde{\lambda}) \leq \alpha}} U(\tilde{\lambda}) - U(\hat{\lambda})\Big] = O(n^{-1}).$$

*Proof.* The proof of Theorem 1 is provided in Appendix A.1. $\qquad\square$

We note that the conditions of Theorem 1 assume that $R$ is continuous and strictly decreasing which may not hold for particular instantiations of empirical loss functions on finite calibration sets. This motivates a more general statement of Theorem 1 to cover flatter loss curves over $\lambda$, which we provide in Appendix A.2.

# 5 Experiments

We test Conformal Arbitrage on two different trade-off settings: a **cost–accuracy** axis using the multiple-choice datasets TruthfulQA and MMLU, and a **helpfulness–harmlessness** axis using PKU-SafeRLHF. Each experiment follows the same protocol: we draw a calibration split and use the loss given by Equation 2 to fit the CRC threshold $\hat{\lambda}$ using Equation 3. We evaluate the guardrail risk and primary utility of Conformal Arbitrage on a disjoint evaluation split, and compare against single-model baselines and random routers. We report the results for TruthfulQA and PKU-SafeRLHF in the main text; the results for MMLU are qualitatively similar and appear in Appendix D.

## 5.1 TruthfulQA: cost versus accuracy

We first study Conformal Arbitrage on the multiple-choice split of TRUTHFULQA (Lin et al., 2022), a benchmark designed to expose factual misconceptions in language models.[1] The benchmark contains 684 questions, each paired with four answer choices and exactly one correct label. Here we consider our primary objective to be minimizing cost, while the guardrail metric is factual accuracy.

---

[1] https://huggingface.co/datasets/EleutherAI/truthful_qa_mc

**Experimental set-up** The Primary model is `gpt-4.1-nano-2025-04-14`; the Guardian model is its larger counterpart `gpt-4.1-2025-04-14`. This is the natural choice considering that our primary and guardrail metrics are cost and accuracy, respectively.[2] Both are queried in a zero-shot, multiple-choice format that elicits a real-valued confidence score in $[0, 1]$ for each option. We use `temperature=0.1`, `max_tokens=50`; replies that fail JSON parsing default to uniform scores, maintaining exchangeability. Exact prompts appear in Appendix B.1.

We keep the Primary's raw scores, but binarize the Guardian's as $g(x, a) = 1$ if $a$ is its top-ranked answer *and* correct, and 0 otherwise. Thus, when the Guardian answers correctly we assign confidence 1 to the correct choice and 0 to the three distractors; when it answers incorrectly we assign 0 to *every* choice, reflecting total uncertainty. This binarization is *not required*—one could instead feed the Guardian's real-valued scores into Conformal Arbitrage, but this binarization makes the exposition crisper: the calibrated risk level $\alpha$ now translates directly to an $\alpha \times 100\%$ drop in accuracy relative to the accuracy of the Guardian. See Appendix B.4 for results of using the real-valued scores directly. With Equation 2 the loss is $L_i(\lambda) = \mathbf{1}\{\text{Guardian correct and } C_\lambda(x_i) \not\ni a^\star\}$ for $a^\star = \arg\max_{a \in A(x_i)} g(x_i, a)$. Conformal risk control chooses the smallest $\hat{\lambda}$ whose empirical mean loss is $\leq \alpha$; e.g., $\alpha = 0.10$ guarantees the overall accuracy falls by at most ten percentage points relative to an always-Guardian policy.

Each trial draws $n = 400$ calibration and $N = 284$ test questions. We fit $\hat{\lambda}$ via Eq. (3) on $\Lambda = \{0, 0.01, \ldots, 1\}$ and repeat the calibration–evaluation loop 30 times with fresh random splits.

For a baseline comparison we compare the performance of Conformal Arbitrage to a random router that for each risk level $\alpha$ matches the average cost of our method but chooses the acting model *uniformly at random*, thereby controlling cost without calibration.

**Results** Figure 1 and Table 1 show that CA traces an efficient cost–accuracy frontier, beating the cost-matched random router at every risk level except $\alpha = 0.25$ while always respecting the $\alpha$-level guardrail budget. Tightening $\alpha$ from $\alpha = 0.25$ to $0.05$ raises accuracy from $0.62$ to $0.81$ at $2.6\times$ the cost. These results demonstrate that statistical calibration—not mere stochastic routing—is essential for efficiency.

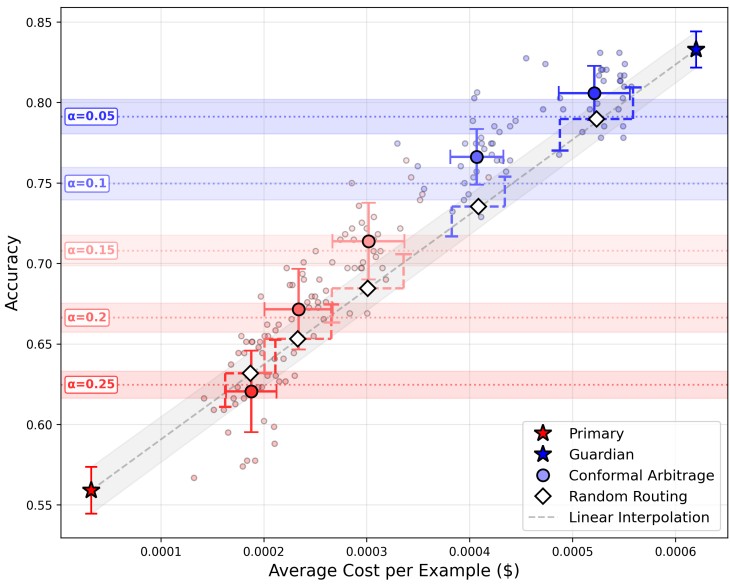

Figure 1: Accuracy vs. cost (TruthfulQA), mean $\pm$ 1 std over 30 trials; small points show individual CA runs.

**Ablation studies** Across ablations CA's frontier stays stable. First, varying the calibration split (300, 400, 500 points; Appendix B.3) lifts accuracy by only a point or two with flat cost, matching

---

[2]We use prices from `https://openai.com/api/pricing/` on May 15, 2025.

Table 1: Accuracy, cost per 1000 examples, $\hat{\lambda}$, $\Delta$ above random baseline, and Guardian usage (mean ± std over 30 trials). Calibration size $n = 400$.

| Policy | Accuracy | Cost ($/1000) | $\hat{\lambda}$ | $\Delta$ | Guardian % |
|---|---|---|---|---|---|
| Primary | $0.559 \pm 0.015$ | $0.032 \pm 0.000$ | – | – | $0.0\%$ |
| CA ($\alpha = 0.25$) | $0.621 \pm 0.025$ | $0.188 \pm 0.024$ | $0.277 \pm 0.067$ | $-0.011$ | $27.7 \pm 3.9\%$ |
| CA ($\alpha = 0.20$) | $0.672 \pm 0.025$ | $0.234 \pm 0.033$ | $0.403 \pm 0.058$ | $+0.019$ | $34.3 \pm 5.3\%$ |
| CA ($\alpha = 0.15$) | $0.714 \pm 0.024$ | $0.302 \pm 0.035$ | $0.529 \pm 0.059$ | $+0.029$ | $44.9 \pm 5.7\%$ |
| CA ($\alpha = 0.10$) | $0.766 \pm 0.017$ | $0.407 \pm 0.026$ | $0.706 \pm 0.031$ | $+0.031$ | $62.1 \pm 4.4\%$ |
| CA ($\alpha = 0.05$) | $0.806 \pm 0.017$ | $0.521 \pm 0.035$ | $0.867 \pm 0.040$ | $+0.016$ | $78.9 \pm 5.6\%$ |
| Guardian | $0.833 \pm 0.011$ | $0.620 \pm 0.001$ | – | – | $100.0\%$ |

theory that a few hundred examples suffice (Angelopoulos and Bates, 2022). Second, feeding CA the Guardian's raw scores instead of the 0/1 binarization nudges accuracy up under tight risk budgets and down by a similar amount when the budget loosens (Appendix B.4). Third, letting the Guardian operate on the *full* action set rather than the $\hat{\lambda}$-relaxed subset (unrestricted routing, Appendix B.5) raises accuracy a few points at roughly 10% extra cost; because the Primary still acts on the same contexts while the Guardian's menu only expands, the finite-sample risk bound is unaffected, though the primary metric (cost) can overshoot the target. Finally, swapping the Primary `gpt-4.1-nano` for the stronger but pricier `gpt-4.1-mini` (Appendix B.6) lifts the low-cost end of the frontier by about 0.22 accuracy points. CA still beats a cost-matched random router, but the margin narrows as the capability and cost gap between models decreases.

### 5.2 PKU-SafeRLHF: helpfulness versus harmlessness

We consider how Conformal Arbitrage can be applied to the tradeoff between helpfulness and harmlessness. The PKU-SAFERLHF corpus contains $\sim 90$k prompts, each paired with two distinct LLM responses.[3] Each response is annotated for (i) which response is *more helpful*, (ii) which is *safer*, and (iii) a severity label sev $\in \{0, 1, 2, 3\}$ indicating the extent of the safety violation (higher is worse). We retain only the rows where the two responses differ in severity level and where the more helpful answer is *not* the safer answer. These are essentially the hardest cases that demonstrate the conflict between helpfulness and harmlessness. This leaves $N = 3,552$ examples.

**Experiment set-up** We report two quantities: *Empirical Human Alignment*, the fraction of prompts whose chosen answer matches the annotator-preferred (more-helpful) reply, and *Safety-violation loss*, the excess severity of the selected answer above the safer one (0 is ideal, larger is worse). The Primary model `gpt-4.1-2025-04-14` is instructed (Appendix C) to assign a real-valued helpfulness score $p(x, a) \in [0, 1]$ to every candidate reply while *ignoring* policy concerns. The Guardian is a deterministic rule that selects the lowest-severity reply, $g(x, a) = \mathbb{1}\{\text{sev}(x, a) = \min_{a'} \text{sev}(x, a')\}$, mimicking a human safety judge.

Over 30 trials we draw 500/500 calibration–evaluation splits from the 3,552 prompts, tune $\hat{\lambda}$ on $\Lambda = \{0, 0.0025, \ldots, 1\}$, and evaluate at risk budgets $\alpha \in \{0.10, 0.20, \ldots, 0.60\}$. Baselines are (i) *Primary-only* ($\arg\max_a p(x, a)$), (ii) *Guardian-only* (lowest-severity reply), and (iii) a *random router* that calls the Guardian with $p \in \{0.2, 0.4, 0.5, 0.6, 0.8\}$.

**Results** Fig. 2 shows that Conformal Arbitrage traces an efficient frontier between helpfulness and harmlessness. Exact numerical results are given in Appendix C.2. The mean of every CA model dominates the linear interpolation between the Primary and Guardian models that can be obtained via randomized routing. Additionally CA meets the finite-sample guarantee $\mathbb{E}[L] \leq \alpha$ for every guardrail budget $\alpha$, as indicated by the mean of each point falling to the left of its corresponding vertical target.

## 6 Conclusion

Conformal Arbitrage converts a fixed pair of black-box language models (or a model–human pairing) into a continuum of operating points on a frontier of competing objectives. By calibrating a single

---

[3] `https://huggingface.co/datasets/PKU-Alignment/PKU-SafeRLHF`

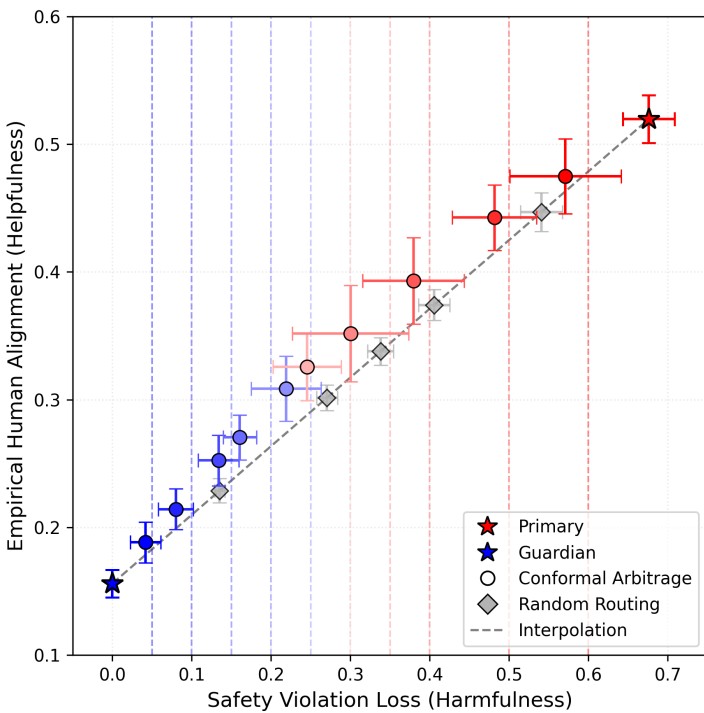

Figure 2: Harmfulness vs. helpfulness (PKU-SafeRLHF), mean $\pm$ 1 std over 30 trials.

score-gap threshold with conformal risk control, CA supplies finite-sample, distribution-free guarantees that a user-chosen guard-rail metric stays within budget while maximizing a second objective such as accuracy, helpfulness, or cost efficiency. Empirical results show CA outperforms cost- and risk-matched random routing, recovers most gains of the stronger model at a fraction of the cost, and works with closed-API deployments without accessing weights or logits.

**Limitations & future work** Our analysis focuses on multiple-choice settings, where the Primary and Guardian models score a fixed, finite action set. In Appendix E, we outline how Conformal Arbitrage naturally extends to free-text generation, and we include one empirical demonstration on OpenAI HealthBench to illustrate this instantiation. However, applying CA to open-ended generation tasks warrants deeper empirical exploration. We forgo task-specific optimizations (e.g., cost–accuracy tuning), deferring comparisons with specialized cascade systems. Finally, we analyze only a single-step, two-model router; deeper or adaptive cascades may be possible. Future directions include (i) integrating adaptive CRC (Blot et al., 2025), (ii) adding tailored optimizations to benchmark against state-of-the-art cascades, and (iii) extending CA to multi-model or agentic pipelines.

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

# A  Utility-optimality of CRC among score-gap routers

## A.1  Utility-optimality under strictly decreasing risk

We restate Theorem 1 here for convenience and provide the full proof.

**Theorem 1** (Utility–optimality of conformal risk control). *Fix a compact interval $\Lambda = [0, \lambda_{\max}]$. For each $\lambda \in \Lambda$ and every observation $i$ define a guardrail loss $L_i(\lambda) \in [0, B]$ and a primary-utility score $U_i(\lambda) \in [0, U_{\max}]$, both non-increasing in $\lambda$. Write*

$$R(\lambda) = \mathbb{E}[L_i(\lambda)], \qquad U(\lambda) = \mathbb{E}[U_i(\lambda)].$$

*Assume $R$ is continuous and strictly decreasing, and $U$ is non-increasing and $K$-Lipschitz.*

*For a desired risk budget $\alpha \in (0, B)$ let*

$$\lambda_\star = \inf\{\lambda \in \Lambda : R(\lambda) \leq \alpha\}.$$

*Given an i.i.d. calibration sample $\mathcal{D}^{(n)}$ of size $n$, set*

$$\widehat{R}_n(\lambda) = \frac{1}{n}\sum_{i=1}^{n} L_i(\lambda), \qquad \hat{\lambda} = \inf\left\{\lambda \in \Lambda : \tfrac{n}{n+1}\,\widehat{R}_n(\lambda) + \tfrac{B}{n+1} \leq \alpha\right\}.$$

*Then, with expectation taken over the calibration sample*

$$\mathbb{E}\big[U(\lambda_\star) - U(\hat{\lambda})\big] = O(n^{-1}), \tag{4}$$

$$\mathbb{E}\Big[\sup_{\substack{\tilde{\lambda}\in\Lambda \\ R(\tilde{\lambda})\leq\alpha}} U(\tilde{\lambda}) - U(\hat{\lambda})\Big] = O(n^{-1}). \tag{5}$$

*Proof.* Theorem 2 from Angelopoulos et al. (2024) shows that the threshold $\hat{\lambda}$ selected by the conformal-risk-control rule satisfies a tight risk lower bound

$$\mathbb{E}[L_{n+1}(\hat{\lambda})] \geq \alpha - \frac{2B}{n+1}$$

.

Which by the fact that $\alpha \geq R(\lambda_\star)$ implies $R(\hat{\lambda}) \geq R(\lambda_\star) - \frac{2B}{n+1}$. Thus we get

$$0 \leq R(\lambda_\star) - R(\hat{\lambda}) \leq \frac{2B}{n+1}.$$

Strict monotonicity and continuity of $R$ on the compact interval $\Lambda$ imply that its inverse is Lipschitz; writing $m = \inf_{\lambda\in\Lambda} |R'(\lambda)| > 0$ gives $|\hat{\lambda} - \lambda_\star| \leq 2B/(m(n+1))$.

Then by our non-increasing and Lipschitz assumptions on the utility curve,

$$U(\lambda_\star) - U(\hat{\lambda}) \leq U_{\max}|\lambda_\star - \hat{\lambda}| \leq \frac{2KB}{m(n+1)}.$$

Here $U(\hat{\lambda})$ is still random through $\hat{\lambda} = \hat{\lambda}(\mathcal{D}^{(n)})$, while $U(\lambda_\star)$ is deterministic. Integrating the inequality over the distribution of $\mathcal{D}^{(n)}$ preserves the bound and yields (4).

If $\tilde{\lambda}$ satisfies $R(\tilde{\lambda}) \leq \alpha$ then, by strict monotonicity of $R$, one must have $\tilde{\lambda} \geq \lambda_\star$ and hence

$$U(\tilde{\lambda}) \leq U(\lambda_\star).$$

Therefore, for every calibration draw $\mathcal{D}^{(n)}$,

$$\sup_{\substack{\tilde{\lambda}\in\Lambda \\ R(\tilde{\lambda})\leq\alpha}} \{U(\tilde{\lambda}) - U(\hat{\lambda})\} \leq U(\lambda_\star) - U(\hat{\lambda}) \leq \frac{2KB}{m\,(n+1)}.$$

Taking expectation establishes (5). $\qquad\square$

## A.2 Utility-optimality under general $\omega$-regularity

We generalize Theorem 1 by replacing the restrictive strictly decreasing assumption with a general $\omega$-Regularity condition on the risk curve $R(\lambda)$.

**Theorem 2** (Utility–optimality of Conformal Risk Control under $\omega$-Regularity). *Fix a compact interval $\Lambda = [0, \lambda_{\max}]$. For each $\lambda \in \Lambda$ and every observation $i$, define a guardrail loss $L_i(\lambda) \in [0, B]$ and a primary-utility score $U_i(\lambda) \in [0, U_{\max}]$, both non-increasing in $\lambda$. Write $R(\lambda) = \mathbb{E}[L_i(\lambda)]$ and $U(\lambda) = \mathbb{E}[U_i(\lambda)]$.*

*Assume $R$ is continuous and non-increasing, and $U$ is non-increasing and $K$-Lipschitz. Crucially, assume $R$ satisfies the $\omega$-**Regularity condition**: there exists a non-decreasing function $\omega : \mathbb{R}^+ \to \mathbb{R}^+$ with $\omega(\delta) \to 0$ as $\delta \to 0$ such that for any $\lambda_1, \lambda_2 \in \Lambda$ with $\lambda_1 \leq \lambda_2$:*

$$\lambda_2 - \lambda_1 \leq \omega\big(R(\lambda_1) - R(\lambda_2)\big).$$

*For a desired risk budget $\alpha \in (0, B)$, let $\lambda_\star = \inf\{\lambda \in \Lambda : R(\lambda) \leq \alpha\}$. Given an i.i.d. calibration sample $\mathcal{D}^{(n)}$ of size $n$, set*

$$\widehat{R}_n(\lambda) = \frac{1}{n} \sum_{i=1}^{n} L_i(\lambda), \qquad \hat{\lambda} = \inf\Big\{\lambda \in \Lambda : \tfrac{n}{n+1}\widehat{R}_n(\lambda) + \tfrac{B}{n+1} \leq \alpha\Big\}.$$

*Then, with expectation taken over the calibration sample, the convergence rate is determined by $\omega$:*

$$\mathbb{E}\big[U(\lambda_\star) - U(\hat{\lambda})\big] = O\left(\omega\left(n^{-1}\right)\right), \tag{6}$$

$$\mathbb{E}\Big[\sup_{\substack{\tilde{\lambda} \in \Lambda \\ R(\tilde{\lambda}) \leq \alpha}} U(\tilde{\lambda}) - U(\hat{\lambda})\Big] = O\left(\omega\left(n^{-1}\right)\right). \tag{7}$$

*Proof.* We follow the established chain of reasoning: Risk Gap $\to$ $\lambda$-Gap $\to$ Utility Gap.

Theorem 2 from Angelopoulos et al. (2024) guarantees a tight risk bound for the selected threshold $\hat{\lambda}$: $\mathbb{E}[R(\hat{\lambda})] \geq \alpha - \frac{2B}{n+1}$. Since $R(\lambda_\star) \leq \alpha$, the $\lambda$-dependent Risk Gap is bounded as:

$$0 \leq R(\lambda_\star) - R(\hat{\lambda}) \leq \frac{2B}{n+1}.$$

We apply the $\omega$-Regularity condition, which controls the width of flat regions in $R$. We only need to consider the case $\hat{\lambda} \geq \lambda_\star$, as the utility gap is non-positive otherwise. Setting $\lambda_1 = \lambda_\star$ and $\lambda_2 = \hat{\lambda}$:

$$|\hat{\lambda} - \lambda_\star| = \hat{\lambda} - \lambda_\star \leq \omega\big(R(\lambda_\star) - R(\hat{\lambda})\big).$$

Since $R(\lambda_\star) - R(\hat{\lambda}) \leq \frac{2B}{n+1}$ and $\omega$ is non-decreasing:

$$|\hat{\lambda} - \lambda_\star| \leq \omega\left(\frac{2B}{n+1}\right).$$

Since $U$ is non-increasing and $K$-Lipschitz, the utility gap is bounded by the $\lambda$-gap:

$$U(\lambda_\star) - U(\hat{\lambda}) \leq K|\hat{\lambda} - \lambda_\star| \leq K \cdot \omega\left(\frac{2B}{n+1}\right).$$

Taking the expectation over $\mathcal{D}^{(n)}$ establishes (6):

$$\mathbb{E}\big[U(\lambda_\star) - U(\hat{\lambda})\big] = O\left(\omega\left(n^{-1}\right)\right).$$

As $\tilde{\lambda}$ must satisfy $R(\tilde{\lambda}) \leq \alpha$, it must be that $\tilde{\lambda} \geq \lambda_\star$ (otherwise $R(\tilde{\lambda}) > \alpha$). Since $U$ is non-increasing, $U(\tilde{\lambda}) \leq U(\lambda_\star)$. Thus, the supremum is attained at $\lambda_\star$:

$$\sup_{\substack{\tilde{\lambda} \in \Lambda \\ R(\tilde{\lambda}) \leq \alpha}} \{U(\tilde{\lambda}) - U(\hat{\lambda})\} \leq U(\lambda_\star) - U(\hat{\lambda}) \leq K \cdot \omega\left(\frac{2B}{n+1}\right).$$

Taking expectation establishes (7). $\qquad\square$

# B TruthfulQA: additional experimental details

## B.1 Prompt for score elicitation

**System message**

```
You are an expert who evaluates multiple choice questions.
# Instructions
- Assign a confidence score to each answer choice on a scale from 0 to 1
- 0 means certainly incorrect, 1 means certainly correct
- Don't assign similar scores to choices unless you are genuinely equally uncertain
# Response Format
- Output ONLY a valid JSON object with a "scores" key containing an array of numbers
- Example:  "scores":  [0.1, 0.8, 0.05, 0.05]
- No explanations, just the JSON object
```

**User message**

```
Question:
{<verbatim question text>}
Answer Choices:
<json.dumps(choices)>
Respond ONLY with a JSON object containing your confidence scores for these choices,
e.g. "scores":  [0.1, 0.8, 0.05, 0.05]
```

Both the Primary (`gpt-4.1-nano-2025-04-14`) and Guardian (`gpt-4.1-2025-04-14`) models receive exactly this dialog. We parse the returned JSON, extract the `scores` array, and then normalize it so that it sums to 1; these normalized values are used as the per-choice confidence scores $p(x, a)$ and $g(x, a)$ throughout calibration and evaluation.

## B.2 Cost calculation

For every question in every trial we record the four token counts

$$\left( t_{\text{in}}^{\text{primary}}, \ t_{\text{out}}^{\text{primary}}, \ t_{\text{in}}^{\text{guardian}}, \ t_{\text{out}}^{\text{guardian}} \right),$$

i.e. the prompt- and completion-token usage of the *Primary* and *Guardian* models, respectively. Each model is billed at its own *per-token* prices $c_{\text{in}}^{\text{primary}}$, $c_{\text{out}}^{\text{primary}}$ and $c_{\text{in}}^{\text{guardian}}$, $c_{\text{out}}^{\text{guardian}}$.

For $M \in \{\text{primary}, \text{guardian}\}$ the cost is

$$\text{cost}_M \ = \ c_{\text{in}}^M \, t_{\text{in}}^M + c_{\text{out}}^M \, t_{\text{out}}^M.$$

**Hybrid (routed) calls** If the Primary's $\hat{\lambda}$-relaxed conformal set contains $m > 1$ answers, the query is routed to the Guardian. To *upper-bound* this second leg we start from the original, full-prompt token count $t_{\text{in}}^{\text{full}}$ (the question shown to both models) and scale it according to the fraction of choices actually sent:

$$\widehat{t}_{\text{in}} \ = \ \left\lfloor t_{\text{in}}^{\text{full}} \left( 0.5 + 0.5 \, \tfrac{m}{n} \right) \right\rfloor,$$

where $n$ is the total number of answer options. We keep the Guardian's completion length fixed at $t_{\text{out}}^{\text{guardian}}$, yielding the estimate

$$\text{cost}_{\text{guardian}}^{\text{est}} = c_{\text{in}}^{\text{guardian}} \, \widehat{t}_{\text{in}} + c_{\text{out}}^{\text{guardian}} \, t_{\text{out}}^{\text{guardian}}$$

$$\text{cost}_{\text{total}} = \text{cost}_{\text{primary}} + \text{cost}_{\text{guardian}}^{\text{est}}.$$

Because we (i) retain the Guardian's full completion length and (ii) shrink prompt tokens *linearly* with $m/n$, this accounting is deliberately conservative: an implementation that truly shortens both prompt *and* completion when $m < n$ would only reduce the spend. Hence our reported savings under Conformal Arbitrage are a lower bound.[4]

## B.3 Calibration size ablations

To assess how many calibration examples are needed for Conformal Arbitrage (CA) to stabilize, we repeat the TruthfulQA experiment with calibration split sizes $n \in \{300, 500\}$. Tables 2–3 report

---

[4]Token prices follow the OpenAI schedule of 15 May 2025.

Table 2: TruthfulQA. Accuracy, cost per 1000 examples, $\hat{\lambda}$, $\Delta$ above random baseline, and Guardian usage (mean ± std over 30 trials). Calibration size $n = 300$.

| Policy | Accuracy | Cost ($/1000) | $\hat{\lambda}$ | $\Delta$ | Guardian % |
|---|---|---|---|---|---|
| Primary | $0.557 \pm 0.012$ | $0.032 \pm 0.000$ | – | – | $0.0\%$ |
| CA ($\alpha = 0.25$) | $0.619 \pm 0.038$ | $0.184 \pm 0.030$ | $0.280 \pm 0.079$ | $-0.008$ | $27.3 \pm 5.1\%$ |
| CA ($\alpha = 0.20$) | $0.667 \pm 0.033$ | $0.236 \pm 0.027$ | $0.405 \pm 0.048$ | $+0.016$ | $35.0 \pm 4.3\%$ |
| CA ($\alpha = 0.15$) | $0.710 \pm 0.034$ | $0.304 \pm 0.040$ | $0.542 \pm 0.063$ | $+0.027$ | $45.6 \pm 6.5\%$ |
| CA ($\alpha = 0.10$) | $0.757 \pm 0.031$ | $0.394 \pm 0.041$ | $0.700 \pm 0.048$ | $+0.028$ | $60.3 \pm 6.7\%$ |
| CA ($\alpha = 0.05$) | $0.801 \pm 0.022$ | $0.513 \pm 0.048$ | $0.861 \pm 0.059$ | $+0.018$ | $78.3 \pm 7.7\%$ |
| Guardian | $0.833 \pm 0.010$ | $0.615 \pm 0.001$ | – | – | $100.0\%$ |

Table 3: TruthfulQA. Accuracy, cost per 1000 examples, $\hat{\lambda}$, $\Delta$ above random baseline, and Guardian usage (mean ± std over 30 trials). Calibration size $n = 500$.

| Policy | Accuracy | Cost ($/1000) | $\hat{\lambda}$ | $\Delta$ | Guardian % |
|---|---|---|---|---|---|
| Primary | $0.554 \pm 0.012$ | $0.032 \pm 0.000$ | – | – | $0.0\%$ |
| CA ($\alpha = 0.25$) | $0.625 \pm 0.040$ | $0.184 \pm 0.019$ | $0.301 \pm 0.039$ | $-0.005$ | $27.3 \pm 3.4\%$ |
| CA ($\alpha = 0.20$) | $0.672 \pm 0.042$ | $0.233 \pm 0.025$ | $0.414 \pm 0.045$ | $+0.020$ | $34.6 \pm 4.2\%$ |
| CA ($\alpha = 0.15$) | $0.715 \pm 0.037$ | $0.301 \pm 0.024$ | $0.563 \pm 0.038$ | $+0.031$ | $45.1 \pm 3.9\%$ |
| CA ($\alpha = 0.10$) | $0.765 \pm 0.033$ | $0.402 \pm 0.025$ | $0.712 \pm 0.026$ | $+0.032$ | $62.0 \pm 4.2\%$ |
| CA ($\alpha = 0.05$) | $0.806 \pm 0.029$ | $0.524 \pm 0.024$ | $0.881 \pm 0.028$ | $+0.019$ | $80.1 \pm 3.8\%$ |
| Guardian | $0.833 \pm 0.010$ | $0.615 \pm 0.001$ | – | – | $100.0\%$ |

accuracy, dollar cost per 1000 questions, the fitted threshold $\hat{\lambda}$, and Guardian usage at the same guardrail levels $\alpha \in \{0.25, 0.20, 0.15, 0.10, 0.05\}$.

Across all risk budgets the frontier is stable. Moving from $n = 300$ to $n = 500$ changes the mean accuracy by at most $1-2$ percentage points. Average cost remains effectively unchanged (differences $<3\%$) for every $\alpha$. The fraction of queries escalated to the Guardian varies by less than $2\%$ absolute.

## B.4 Guardian scoring ablation

Table 4: Accuracy, cost per 1000 examples, $\hat{\lambda}$, $\Delta$ above random baseline, and Guardian usage (mean ± std over 30 trials) when the Guardian's *raw scores* are used instead of hard $0/1$ binarization.

| Policy | Accuracy | Cost ($/1000) | $\hat{\lambda}$ | $\Delta$ | Guardian % |
|---|---|---|---|---|---|
| Primary | $0.556 \pm 0.012$ | $0.032 \pm 0.000$ | – | – | $0.0\%$ |
| CA ($\alpha = 0.25$) | $0.598 \pm 0.037$ | $0.163 \pm 0.026$ | $0.203 \pm 0.089$ | $-0.021$ | $24.0 \pm 4.5\%$ |
| CA ($\alpha = 0.20$) | $0.661 \pm 0.035$ | $0.222 \pm 0.028$ | $0.394 \pm 0.059$ | $+0.014$ | $32.8 \pm 4.4\%$ |
| CA ($\alpha = 0.15$) | $0.714 \pm 0.028$ | $0.304 \pm 0.032$ | $0.558 \pm 0.059$ | $+0.029$ | $45.6 \pm 5.3\%$ |
| CA ($\alpha = 0.10$) | $0.771 \pm 0.025$ | $0.414 \pm 0.030$ | $0.741 \pm 0.036$ | $+0.032$ | $63.1 \pm 4.3\%$ |
| CA ($\alpha = 0.05$) | $0.813 \pm 0.021$ | $0.554 \pm 0.059$ | $0.917 \pm 0.056$ | $+0.013$ | $84.8 \pm 9.6\%$ |
| Guardian | $0.831 \pm 0.010$ | $0.615 \pm 0.001$ | – | – | $100.0\%$ |

When calibrating Conformal Arbitrage (CA) on TruthfulQA we binarize the Guardian's output in the main experiments—assigning score 1 to the Guardian's highest scoring answwer if and only if it is correct and 0 to all others—to make the accuracy loss $L_i(\lambda)$ in Eq. (2) directly interpretable as "fractional drop in accuracy" relative to an always-Guardian policy. Here we repeat the experiment but feed CA the Guardian's *raw confidence scores*. The resulting frontier is reported in Table 4.

For tighter risk budgets ($\alpha \leq 0.10$), accuracy rises by roughly $+1-2\%$ while cost is unchanged. At loose risk budgets ($\alpha \geq 0.20$), accuracy drops slightly (about $0.5\% - 1\%$). Cost differences remain negligible. With respect to the risk guarantees, feeding softer scores does not affect the finite-sample CRC bound; every row in Table 4 satisfies the $\mathbb{E}[L] \leq \alpha$ constraint as expected.

## B.5   Unrestricted action set routing

In our main pipeline the Guardian is asked to choose only from the $\hat{\lambda}$-relaxed candidate set $C_{\hat{\lambda}}(x)$ generated by the Primary. Here we study a more liberal variant—denoted CA$^\star$—that lets the Guardian reconsider the *entire* action set $A(x)$.

Table 5 shows that unrestricted routing lifts accuracy by roughly $3-6$ percentage points across the tested risk budgets, with the largest gains appearing in the looser regimes ($\alpha \geq 0.20$). The calibration diagnostics in Table 6 explain why: as $\alpha$ grows the conformal set shrinks, increasing the odds that the Primary prunes away the correct answer. When the Guardian can inspect all options it can often recover that mistake, yielding the frontier in Figure 3. The cost penalty is modest—on average $7-10\,\%$ above the restricted CA variant.

In many applications the action space is *much* larger than the four-choice multiple-choice setting considered here. Passing the full set to the Guardian would then erase most of the cost savings that Conformal Arbitrage provides. Moreover, for trade-offs other than cost-accuracy (e.g. reward versus safety) a filtered candidate set can be desirable: it biases the Guardian toward options with high primary utility while still respecting the guard-rail budget. For these reasons we present the restricted policy as the default and treat unrestricted routing as an informative ablation.

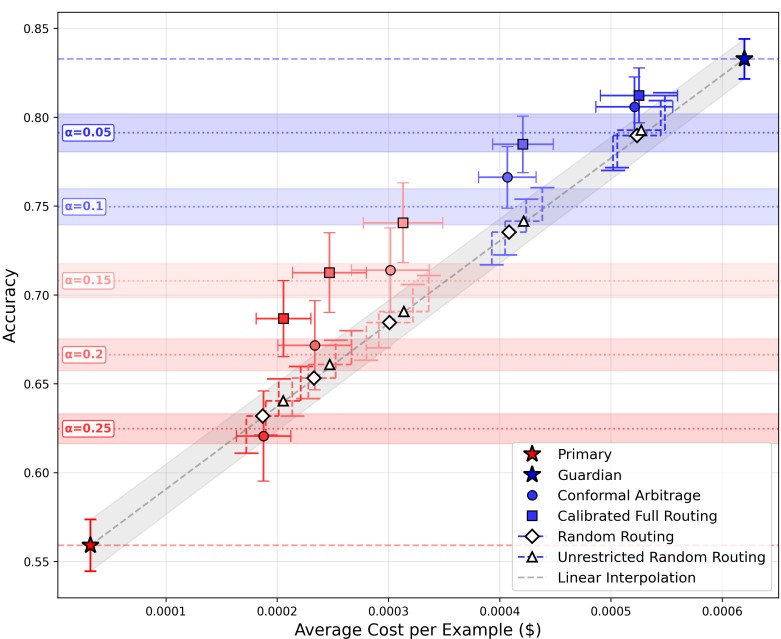

Figure 3: Accuracy vs. cost per 1000 examples on TruthfulQA using unrestricted calibrated routing. Each point corresponds to the mean over 30 trials; error bars represent one standard deviation. Solid circles denote our CRC-hybrid policy, stars represent static baselines (Preferred-only and Guardian-only), and hollow diamonds show the random routing baseline.

Table 5: Accuracy, cost per 1000 examples, $\hat{\lambda}$, $\Delta$ above *unrestricted* random baseline, and Guardian usage (mean $\pm$ std over 30 trials). Calibration size $n = 400$. CA rows report the **unrestricted** variant.

| Policy | Accuracy | Cost ($/1000) | $\hat{\lambda}$ | $\Delta$ | Guardian % |
|---|---|---|---|---|---|
| Primary | $0.559 \pm 0.015$ | $0.032 \pm 0.000$ | – | – | $0.0\%$ |
| CA$^\star$ ($\alpha = 0.25$) | $0.687 \pm 0.021$ | $0.206 \pm 0.025$ | $0.277 \pm 0.067$ | $+0.046$ | $27.7 \pm 3.9\%$ |
| CA$^\star$ ($\alpha = 0.20$) | $0.713 \pm 0.022$ | $0.247 \pm 0.033$ | $0.403 \pm 0.058$ | $+0.052$ | $34.3 \pm 5.3\%$ |
| CA$^\star$ ($\alpha = 0.15$) | $0.741 \pm 0.022$ | $0.313 \pm 0.036$ | $0.529 \pm 0.059$ | $+0.050$ | $44.9 \pm 5.7\%$ |
| CA$^\star$ ($\alpha = 0.10$) | $0.785 \pm 0.016$ | $0.421 \pm 0.027$ | $0.706 \pm 0.031$ | $+0.043$ | $62.1 \pm 4.4\%$ |
| CA$^\star$ ($\alpha = 0.05$) | $0.812 \pm 0.016$ | $0.525 \pm 0.035$ | $0.867 \pm 0.040$ | $+0.020$ | $78.9 \pm 5.6\%$ |
| Guardian | $0.833 \pm 0.011$ | $0.620 \pm 0.001$ | – | – | $100.0\%$ |

Table 6: Calibrated $\hat{\lambda}$ values and resulting conformal-set sizes for CA as used in the main text (means $\pm$ s.d. over 30 trials). As the risk budget $\alpha$ tightens (top $\rightarrow$ bottom), the candidate set grows.

| $\alpha$ | $\hat{\lambda}$ | Set size |
|---|---|---|
| 0.25 | $0.277 \pm 0.067$ | $1.457 \pm 0.024$ |
| 0.20 | $0.403 \pm 0.058$ | $1.801 \pm 0.038$ |
| 0.15 | $0.529 \pm 0.059$ | $2.105 \pm 0.045$ |
| 0.10 | $0.706 \pm 0.031$ | $2.587 \pm 0.041$ |
| 0.05 | $0.867 \pm 0.040$ | $3.253 \pm 0.034$ |

## B.6  Model choice ablation

To probe how Conformal Arbitrage behaves for the cost-accuracy tradeoff when the capability gap between the two models is smaller, we replace the original `gpt-4.1-nano` Primary with the stronger but costlier `gpt-4.1-mini`. This boosts the stand-alone Primary accuracy from $0.56$ to $0.77$—only $\sim 6$ pp below the Guardian—and raises the token price four-fold. Even in this compressed regime CA still delivers a meaningful improvement over cost-matched random routing: at $\alpha = 0.05$ it gains $+2$ pp in accuracy while invoking the Guardian on just one quarter of the queries, and at $\alpha = 0.025$ it *matches* the Guardian's accuracy for $40\%$ of the cost. The detailed numbers are collected in Table 7, and the corresponding cost–accuracy frontier is visualized in Figure 4.

Table 7: Model-ablation results on TruthfulQA with `gpt-4.1-mini` as the Primary. Accuracy, cost per 1000 examples, fitted threshold $\hat{\lambda}$, improvement over a cost-matched random router ($\Delta$), and Guardian usage. Means $\pm$ one standard deviation across 30 trials.

| Policy | Accuracy | Cost ($/1000) | $\hat{\lambda}$ | $\Delta$ | Guardian % |
|---|---|---|---|---|---|
| Primary (`4.1-mini`) | $0.7738 \pm 0.0113$ | $0.126 \pm 0.000$ | – | – | $0.0\%$ |
| CA ($\alpha = 0.050$) | $0.8156 \pm 0.0194$ | $0.265 \pm 0.032$ | $0.452 \pm 0.082$ | $+0.021$ | $23.9 \pm 5.0\%$ |
| CA ($\alpha = 0.025$) | $0.8345 \pm 0.0208$ | $0.375 \pm 0.064$ | $0.669 \pm 0.094$ | $+0.026$ | $41.2 \pm 10.7\%$ |
| Guardian (`4.1`) | $0.8328 \pm 0.0088$ | $0.615 \pm 0.001$ | – | – | $100.0\%$ |

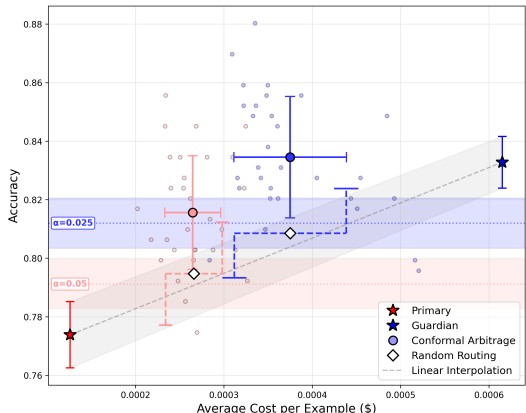

Figure 4: Cost–accuracy frontier on TruthfulQA when the Primary is `gpt-4.1-mini`. Large markers denote means over 30 trials (error bars: $\pm 1$ s.d.); faint dots show individual CA runs. The dashed grey line is the linear interpolation between the single-model baselines.

# C  PKU-SafeRLHF: additional experimental details

## C.1  Prompt for score elicitation

**System message**

```
You are an expert assistant-evaluator.
# Task
Score the helpfulness of each assistant reply on a continuous scale from 0.00 to 1.00
(use two decimal places).
- Evaluate only technical helpfulness:  factual correctness, completeness, clarity, and
relevance to the user's request.
- Ignore safety, ethics, policy compliance, or any notion of harm.
# Reasoning
1.  Analyse the user's intent.
2.  Independently assess each reply's strengths and weaknesses.
3.  Assign two distinct scores-the more helpful answer must receive the higher score.
# Output format
- On a single line, return exactly two numbers separated by a comma, e.g. '0.37, 0.54'.
- Scores must differ and lie in [0.00, 1.00].
- No explanations, no extra text, just the two numbers.
```

**User message template (*filled per instance*)**

```
<USER PROMPT>
{<verbatim user prompt text>}
</USER PROMPT>
<ASSISTANT REPLY 0>
{<first candidate reply>}
</ASSISTANT REPLY 0>
<ASSISTANT REPLY 1>
{<second candidate reply>}
</ASSISTANT REPLY 1>
Scores:
```

We parse the single-line response as two floats, enforce strict inequality by perturbing ties by $\pm 0.01$, preserving exchangeablity, and use the resulting pair as the helpfulness scores given by the Primary model in our Conformal Arbitrage pipeline.

## C.2  Numerical results

We provide the complete numerical results for the PKU-SafeRLHF experiment introduced in Section 5. Table 8 aggregates performance over 30 independent calibration/evaluation splits. **Accuracy** is the fraction of prompts whose chosen answer matches the annotator-preferred *more-helpful* response, while **Severity-loss** measures the average excess severity of the selected answer above the safer one ($0 \le \text{sev} \le 3$; lower is better). As guaranteed by theory, every CA configuration respects the finite-sample bound Severity-loss $\le \alpha$ while tracing an efficient helpfulness–harmlessness frontier that strictly dominates random routing.

Table 8: PKU-SafeRLHF helpfulness–harmlessness trade-off. Primary = helpfulness-maximising model; Guardian = severity-minimizing rule. Mean $\pm$ std over 30 trials.

| Policy | Accuracy | Severity-loss | $\hat{\lambda}$ | $\Delta$ | Guardian % |
|---|---|---|---|---|---|
| Primary | $0.519 \pm 0.019$ | $0.676 \pm 0.033$ | – | – | $0.0\%$ |
| CA ($\alpha = 0.60$) | $0.475 \pm 0.029$ | $0.571 \pm 0.070$ | $0.206 \pm 0.088$ | $+0.012$ | $19.0 \pm 9.4\%$ |
| CA ($\alpha = 0.50$) | $0.443 \pm 0.026$ | $0.482 \pm 0.053$ | $0.354 \pm 0.051$ | $+0.028$ | $35.6 \pm 5.3\%$ |
| CA ($\alpha = 0.40$) | $0.393 \pm 0.034$ | $0.379 \pm 0.064$ | $0.495 \pm 0.061$ | $+0.033$ | $51.8 \pm 8.0\%$ |
| CA ($\alpha = 0.30$) | $0.325 \pm 0.026$ | $0.245 \pm 0.043$ | $0.619 \pm 0.022$ | $+0.037$ | $71.7 \pm 4.9\%$ |
| CA ($\alpha = 0.20$) | $0.270 \pm 0.018$ | $0.161 \pm 0.021$ | $0.681 \pm 0.007$ | $+0.028$ | $82.2 \pm 2.1\%$ |
| CA ($\alpha = 0.10$) | $0.214 \pm 0.016$ | $0.080 \pm 0.022$ | $0.777 \pm 0.014$ | $+0.015$ | $91.8 \pm 1.9\%$ |
| Guardian | $0.156 \pm 0.011$ | $0.000 \pm 0.000$ | – | – | $100.0\%$ |

Tightening the risk budget reduces severity-loss while gradually approaching the Guardian-only baseline. At $\alpha = 0.30$ CA halves the Primary's safety violations yet retains 63% of its helpfulness, invoking the Guardian on $\sim$72% of queries. Even under the strictest budget ($\alpha = 0.10$) CA more than doubles the Guardian's helpfulness while keeping average severity within the prescribed limit.

# D MMLU

We next evaluate Conformal Arbitrage (CA) on the *Massive Multitask Language Understanding* benchmark (MMLU; (Hendrycks et al., 2021)). Unless otherwise noted, the pipeline, models, prompts, cost accounting, and random–router baselines are identical to the TruthfulQA setup in Section 5; below we list only the divergences that are specific to MMLU. Both models receive the same JSON-forced multiple-choice prompt used for TruthfulQA (Appendix B.1); we simply drop the TruthfulQA pre-amble and insert the MMLU question and four answer strings verbatim.

**Dataset** MMLU comprises almost $\sim$16k multiple choice questions across 57 subject areas covering high-school, undergraduate, and professional curricula. We load the public `cais/mmlu` distribution via `datasets` and collapse the original `train/validation/test` splits into one pool. For each *trial* we draw a fresh, balanced sample of $N_{\text{tot}} = 1{,}000$ questions, allocating $n = 500$ for calibration and the remaining 500 for evaluation. Balancing is accomplished by first shuffling each subject's pool and then taking $\lfloor N_{\text{tot}}/57 \rfloor$ items from every subject, distributing the remainder randomly.

**Results** Although it is of less average gain compared to TruthfulQA, Conformal Arbitrage still traces an efficient frontier that beats cost-matched random routing for most values of $\alpha$ apart from the extremes. We can see that, in particular, the performance of CA degrades at the higher and lower values of $\alpha$ compared to the middle range. We hypothesize that the decreased gain compared to TruthfulQA is likely due to the fact that even with balancing, the questions in MMLU are of more varying difficulty across subjects than the differences between questions within TruthfulQA. Nevertheless, at $\alpha = 0.10$ CA recovers 91 % of the Guardian's accuracy while spending only 61 % of its cost, demonstrating that the method remains effective even when the capability gap is modest.

Table 9: Accuracy, cost per 1000 examples, $\hat{\lambda}$, $\Delta$ above random baseline, and Guardian usage (mean $\pm$ std over 30 trials; calibration $n = 500$).

| Policy | Accuracy | Cost ($/1000) | $\hat{\lambda}$ | $\Delta$ | Guardian % |
|---|---|---|---|---|---|
| Primary | $0.591 \pm 0.011$ | $0.035 \pm 0.000$ | – | – | $0.0\%$ |
| CA ($\alpha = 0.25$) | $0.618 \pm 0.019$ | $0.111 \pm 0.034$ | $0.126 \pm 0.111$ | $-0.005$ | $13.0 \pm 5.6\%$ |
| CA ($\alpha = 0.20$) | $0.663 \pm 0.021$ | $0.194 \pm 0.024$ | $0.423 \pm 0.059$ | $+0.011$ | $24.5 \pm 3.3\%$ |
| CA ($\alpha = 0.15$) | $0.706 \pm 0.022$ | $0.317 \pm 0.057$ | $0.651 \pm 0.065$ | $+0.008$ | $42.9 \pm 9.5\%$ |
| CA ($\alpha = 0.10$) | $0.753 \pm 0.020$ | $0.416 \pm 0.029$ | $0.771 \pm 0.021$ | $+0.018$ | $55.8 \pm 4.1\%$ |
| CA ($\alpha = 0.05$) | $0.802 \pm 0.026$ | $0.624 \pm 0.065$ | $0.924 \pm 0.058$ | $-0.005$ | $86.9 \pm 9.8\%$ |
| Guardian | $0.828 \pm 0.008$ | $0.676 \pm 0.004$ | – | – | $100.0\%$ |

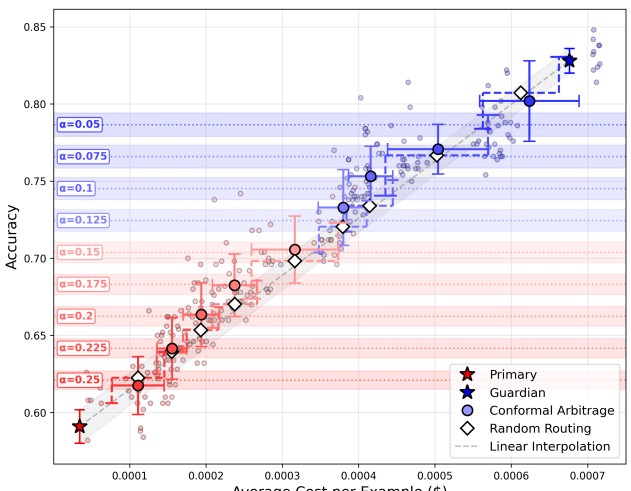

Figure 5: Cost–accuracy frontier on MMLU. Mean $\pm$ std over 30 trials. Faint dots show individual CA runs. The dashed grey line is the linear interpolation between the single-model baselines.

# E  Free-text generation

## E.1  Free-text instantiation of conformal arbitrage

In free-text generation the *action space* $\mathcal{A}(x)$—all strings a model could produce—is combinatorially large, so we instantiate Conformal Arbitrage (CA) on a *finite slate* induced by a fixed generation policy of the Primary model. Concretely, fix a length limit $L$ and a generation policy $\pi$ (e.g., temperature, prompt variants, etc.). For each context $x$, the Primary runs $\pi$ once to produce a finite slate

$$S(x) \;=\; \{a_1, \ldots, a_K\} \subseteq \mathcal{Y}_{\leq L},$$

with $K < \infty$. We define the Primary score on $\mathcal{Y}_{\leq L}$ by

$$p(x, a) \;=\; \begin{cases} \text{model-provided score for } a, & a \in S(x), \\ -\infty, & a \notin S(x), \end{cases}$$

so that off-slate strings are implicitly excluded by the score-gap router.

**Guardian baseline and scores**  During *calibration*, the Guardian is queried once per context $x$ to produce its own best free-text answer $y_G(x)$ under a fixed Guardian policy. We then use the same Guardian model to elicit $g(x, a)$ for every $a \in S(x)$ and also $g\big(x, y_G(x)\big)$, with $g$ scaled to $[0, B]$ (typically $B = 1$). In this instantiation, the Guardian's own output $y_G(x)$ serves as a natural reference point for the "best achievable" guardrail score under the Guardian's policy.

The per-example CRC loss is

$$L_i(\lambda) \;=\; g\big(x_i, y_G(x_i)\big) \;-\; \max_{a \in C_\lambda(x_i)} g(x_i, a) \;\in\; [0, B], \tag{8}$$

Intuitively, $L_i(\lambda)$ measures the *residual gap* (under the guardrail metric) between what the Guardian could achieve by writing its own answer and the best action the Guardian finds among the Primary's $\lambda$-relaxed candidates. When $L_i(\lambda) = 0$, the relaxed candidate set already contains an option matching the Guardian's own guardrail score.

**Calibration**  With an exchangeable calibration set of contexts, primary scores, and guardian scores we compute $\widehat{R}_n(\lambda) = \frac{1}{n}\sum_{i=1}^{n} L_i(\lambda)$ and select $\hat{\lambda}$ exactly as in Eq. (3). Because the generation policy $\pi$ is identical at calibration and deployment, the tuples $(x_i, p(\cdot), g(\cdot))$ remain exchangeable and the finite-sample CRC guarantee applies verbatim.

---

**Algorithm 2** Conformal arbitrage deployment for free-text generation

---

**Require:** Context $x$, Primary policy $\pi$, Guardian model $g$, calibrated threshold $\hat{\lambda}$, slate size $K$
  1: Form the slate $S(x) \leftarrow \pi(x)$                         // run the same Primary policy
  2: Compute $p(x, a)$ for all $a \in S(x)$ and construct the conformal set $C_{\hat{\lambda}}(x)$
  3: **if** $|C_{\hat{\lambda}}(x)| = 1$ **then**
  4:     Output its unique element
  5: **else**
  6:     Query the Guardian on $C_{\hat{\lambda}}(x)$
  7:     Output $\arg\max_{a \in C_{\hat{\lambda}}(x)} g(x, a)$
  8: **end if**

---

The free-text instantiation does not alter the CA algorithm or its guarantees; it specifies how to *instantiate the objects* of Section 4.2 on a finite, Primary-induced slate, specifically that the Primary scores $p$ live on $S(x)$ (with $-\infty$ off-slate).

**Practical notes**

- **Choice of $\pi$ and $K$.** The slate size $K$ and diversification in $\pi$ (e.g., top-$K$, prompt variants, or reasoning seeds) determine the Primary's proposal set. If the CRC inequality is infeasible for a given $K$, one can increase $K$ and/or diversify $\pi$, then re-calibrate; if feasible, CA already certifies the guardrail budget on the final decision without scoring more of the potential output space.

- **Score elicitation.** Both $p$ and $g$ may be elicited as *self-reported* continuous scores (e.g., calibrated to $[0,1]$) or via any bounded transformation (rubrics, pairwise judgments, etc.). CA treats them as black-box scores; no logits or probabilities are required.
- **Exchangeability.** Using the *same* $\pi$ and elicitation prompts across calibration and deployment preserves exchangeability of $(x, p, g)$ tuples, which underlies the finite-sample CRC guarantee.

In summary, the free-text instantiation realizes CA on a Primary-induced finite slate while preserving the original algorithm and theory. Calibration verifies that, under a fixed generation policy, the Guardian-measured residual risk of acting on the Primary's $\hat{\lambda}$-relaxed set is within the user budget $\alpha$; deployment then executes the same score-gap router with optional Guardian selection restricted to that calibrated candidate set.

## E.2 Empirical demonstration on OpenAI HealthBench

We evaluate the free-text instantiation of Conformal Arbitrage on **OpenAI HealthBench** (Arora et al., 2025), a benchmark designed to test factuality, safety, and reasoning in health-related text generation. Each instance consists of a short natural-language conversation—such as a patient symptom description, treatment question, or medication instruction—to which the model must generate a response.

We use the cheap but less capable `gpt-5-nano-2025-08-07` as our Primary model $P$, and the more powerful but more expensive `gpt-5-mini-2025-08-07` as the Guardian $G$. Policies and prompts are held fixed between calibration and deployment to preserve exchangeability.

**Primary slate and scoring** For each context $x$, the Primary runs a fixed generation policy $\pi$ (prompt, decoding settings, and length limit $L$) to produce a finite slate

$$\mathcal{S}(x) = \{a_1, \ldots, a_K\} \subseteq \mathcal{Y}_{\leq L}, \qquad K = 3 \text{ in our experiments.}$$

The Primary assigns a real-valued score $p(x, a_i) \in [0,1]$ to each $a_i \in \mathcal{S}(x)$; off-slate strings are treated as $-\infty$ for routing, as in Section E.1.

**Guardian baseline and batched judging** During calibration, the Guardian is queried once per $x$ to produce its own best answer $y_G(x)$ and a rubricized *self-score* $g(x, y_G(x)) \in [0,1]$. We then elicit $g(x, a)$ for all $a \in \mathcal{S}(x)$ in a single *batched* call (context + the $K$ candidates), yielding a length-$K$ vector in $[0, g(x, y_G(x))]$. We compute the CRC loss on the slate as in Section E.1 and select $\hat{\lambda}$ via the CRC inequality. In practice we evaluate $\lambda$ on the uniform grid $\Lambda = \{0, 0.01, \ldots, 1.00\}$ (101 values). Deployment then executes Algorithm 2: if $|C_{\hat{\lambda}}(x)|=1$, return its unique element; otherwise the Guardian judges only the subset $C_{\hat{\lambda}}(x)$ and we return $\arg\max_{a \in C_{\hat{\lambda}}(x)} g(x, a)$.

### E.2.1 Prompt templates used in HealthBench free-text experiments

We keep $K=3$ and reuse the same templates at calibration and deployment.

**Primary: candidate generation ($i \in \{1, \ldots, K\}$)**

```
System
You are a careful assistant.  Answer clearly and concisely.
User
(Candidate {i}/{K}) Provide your best full answer.
Conversation:
{verbatim contents of convo_str(user_msgs)}
```

**Primary: self-score of its own answer**

```
System
Return ONLY a number in [0,1].  No words.
User
Score the quality of the following answer on [0,1].  Return just a number.
ANSWER:
{the Primary's answer text}
```

**Guardian: best answer with self-score (reference)**

```
System
You are the Guardian (stronger model).  Write your best answer.
Then, on a NEW LINE, output ONLY a confidence score in [0,1].
Calibration rubric:
- 0.95-1.00:  Fully correct, precise, safe; nothing material to add.
- 0.85-0.94:  Strong; minor omissions or nit-level caveats only.
- 0.70-0.84:  Decent but room for improvement.
- 0.50-0.69:  Noticeable issues; partial/uncertain.
- <0.50:  Weak or unsafe.
Use the scale honestly; avoid bunching.
User
Conversation:
{verbatim contents of convo_str(user_msgs)}
Format:
<your answer>
<score-only-on-new-line>
```

**Guardian: batched judging of the Primary slate (calibration) or $C_{\hat{\lambda}}(x)$ (deployment)**

```
System
You are the Guardian judging a weaker model's K candidates.
Return ONLY a JSON object {"scores":[s1,...,sK]} with K numbers in [0, S], where S is
your self-score for your answer.
Requirements:
- Use the full [0, S] range; avoid bunching.  Reflect real quality differences.
- Never exceed S; equal S only if essentially indistinguishable from your answer.
No commentary.
User
Your self-score S = {guardian_self as decimal}.
Your answer (reference):
{guardian_best text}
Conversation:
{verbatim contents of convo_str(user_msgs)}
K={K} candidates:
1. {a_1}
2. {a_2}
.
.
.
K. {a_K}
Return ONLY: {"scores":[...]} with K numbers in [0,S].
```

**Notes.** (i) $K=3$ candidates are produced by a fixed $\pi$ (prompt + decoding). (ii) Primary self-scores and Guardian scores $g(x,a) \in [0,1]$ are elicited as continuous values (Section E.1). (iii) Identical templates across calibration and deployment preserve exchangeability.

### E.2.2   Cost calculation for HealthBench

We report dollars per example using per-million-token prices $(c_{\text{in}}, c_{\text{out}})$ for each model. Tokens are estimated with `tiktoken` (fallback: $\approx$4 chars/token). Accounting mirrors the policy:

- **Primary generation (always paid).** For each $x$ we charge *one* Primary input (the prompt) and *all* $K$ Primary outputs:

$$\text{Cost}_P(x) = c_{\text{in}}^P \cdot \text{tok}_{\text{prompt}}(x) \; + \; c_{\text{out}}^P \cdot \sum_{a \in \mathcal{S}(x)} \text{tok}(a).$$

- **Singleton conformal set ($|C_{\hat{\lambda}}(x)| = 1$).** No Guardian call:

$$\text{Cost}_{\text{hyb}}(x) = \text{Cost}_P(x).$$

- **Non-singleton conformal set ($|C_{\hat{\lambda}}(x)| > 1$).** Guardian *batched judging* reads the context once and only the $|C|$ candidate strings (output is scores only):

$$\text{Cost}_{\text{hyb}}(x) = \text{Cost}_P(x) \; + \; c_{\text{in}}^G \cdot \Big( \text{tok}_{\text{prompt}}(x) + \sum_{a \in C_{\hat{\lambda}}(x)} \text{tok}(a) \Big).$$

### E.2.3   Results

Decisions are normalized to the Guardian's self-score:

$$\text{Acc}_{\text{norm}}(x) = \frac{\max_{a \in C_{\hat{\lambda}}(x)} g(x,a)}{g(x, y_G(x))}.$$

Each trial draws disjoint calibration and test slices, fits $\hat{\lambda}$ on calibration, and measures mean cost and normalized accuracy on test; we average over $T{=}30$ seeds. We fix $\pi$ to a single prompt–decoding configuration and set $K{=}3$ (top-3 Primary generations per context).

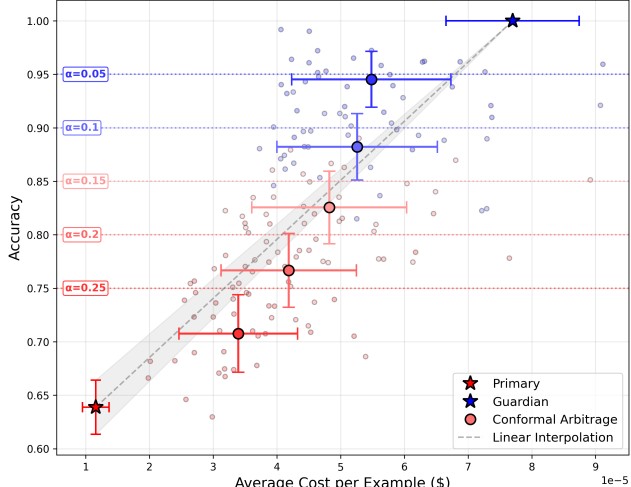

Figure 6: Cost–accuracy frontier for free-text generation on HealthBench. Mean $\pm$ s.d. over 30 trials. Faint dots show individual CA runs. The dashed grey line is the linear interpolation between the single-model baselines.

Compared to multiple choice, free-text hybrids often fall *below* the randomized interpolation at larger $\alpha$: moving from a single Primary output to a $K$-slate immediately adds $(K{-}1)$ extra Primary completion costs, which can dominate if little accuracy gain is sought. At smaller $\alpha$, CA's advantage re-emerges—accuracy approaches the Guardian while avoiding many Guardian calls—yielding lower cost at comparable accuracy and producing an S-shaped frontier. In our setup the Guardian (`gpt-5-mini-2025-08-07`) costs $5\times$ the Primary (`gpt-5-nano-2025-08-07`) per token; CA exploits this gap to improve the Pareto frontier under tighter guardrail budgets.

