# OpenReview forum: "Conformal Arbitrage: Risk-Controlled Balancing of Competing Objectives in Language Models"
_NeurIPS.cc/2025/Conference — NeurIPS 2025 poster_

### Official Review · Reviewer_E6o9 · 2025-07-01

**Clarity:** 2
**Significance:** 2
**Originality:** 2
**Rating:** 4
**Confidence:** 4

**Summary:**

The paper introduces a new method based on conformal risk control for multiple-choice tasks to defer instances from a primary model to a secondary "Guardian" model in order to guarantee a desired safeguard objective while maximizing a primary objective (e.g., accuracy vs. cost). The authors motivate the usage of this method in the context of LLMs and show experimentally that it outperforms random deferral of instances at the same cost.

**Questions:**

- I think it should be made explicitly clear much earlier that the methodological contributions only apply to multiple-choice tasks. This is only stated explicitly in the last paragraph under limitations. In particular, it would make it clearer why in the methodology section the framework is presented in terms of actions and action sets and how you can obtain a score for the actions from the LLM (but this should ideally also be clarified explicitly in this section).
- It seems the methodology and the experiments treat the LLM as a basic classifier (multiple-choice tasks and confidence score output). In this case, could you compare how your method differs from methods in the learning to defer literature?
- Theorem 1 requires $R$ to be continuous and strictly decreasing but to me it seems that both do not hold given the definition of $L_i(\lambda)$ in Equation 2 (Eq. 2 looks like a decreasing piecewise constant function). Could you explain if the theorem would still hold?

**Minor Comments**
- line 53: twice "the the" written
- line 164: "any black-box predictor attains ($1-\alpha$) coverage" this is confusing, the conformal predictor attains the coverage not the black-box classifier.
- line 221: "$\lambda$-relaxed set" should be "$\hat{\lambda}$-relaxed set"?
- Algorithm 1 line 7: "G(a)" should be "g(x,a)"?
- line 241: "$\lambda$ can only shrink the set of contexts" should be "set of actions"?

**Ethical Concerns:**

["NO or VERY MINOR ethics concerns only"]

**Final Justification:**

I think the clarifications discussed during the rebuttal will make the paper more clear and cohesive. The added experiments and instantiation of the CA method for free-text outputs to the appendix will also strengthen the contributions of the paper. Hence, I increased my score to 4. A point that was not quite addressed is if/how CA performance degrades if a guardian model has heterogenous performance across inputs.

I think the paper has the potential to be a clear accept, but without seeing the detailed revisions it is hard to rate it higher.

**Limitations:**

The authors have very briefly addressed limitations. However, I think the main limitation of their method being only applicable in multiple-choice tasks should be mentioned much earlier, such as in the introduction.

**Paper Formatting Concerns:**

No formatting concerns.

**Quality:**

2

**Strengths And Weaknesses:**

**Strengths**
- I think the application of conformal risk control for deferring to specialized (language) models in order to guarantee a certain objective is interesting
- The methodology/theory seems sound to me (up to a minor question see below)
- The experiment setup is in general well chosen for the methodology and the results are convincing, but the random baseline is a weak comparison

**Weaknesses**
- The framework requires specialized models (or a human expert) for each of the objectives. It (implicitly) assumes that in general the specialized model (guardian model) will make less mistakes than the non-specialized one. However, in practice this might not always be the case for all types of inputs, e.g., when the "guardian model" is a human expert. (The experimental setup was chosen such that the assumption holds so it is unclear how beneficial it is when it doesn't.)
- I find the paper lacks coherence in the presentation which makes the paper unclear at times. The introduction and related work section refers broadly to the application of conformal arbitrage to LLMs. However, the methodology section uses the terminology "action" and "action sets" to refer to the output without connecting it to the context of LLMs. The experiments then only handle multiple choice questions with this limitation not being explicitly mentioned earlier.
- the application possibilities/significance in the context of LLMs is narrow due to the method only being applicable to multiple-choice tasks.

---

> ### Author Rebuttal · Authors · 2025-07-31
>
> We thank the reviewer for their engagement with our work and for highlighting the appeal of using conformal risk control (CRC) for black-box deferral. We address the concerns point-by-point below.
>
> Scope beyond multiple choice.
>
> Conformal Arbitrage (CA) is not restricted to multiple choice. The theory in Section 4 is stated for any input-dependent finite action slate $A(x)$; Algorithm 1 and CRC calibration (Eq. (3)) operate on such action sets regardless of how candidates are produced (MCQ options or free-text proposals).
>
> Our limitations sentence 'our study is confined to multiple-choice tasks; applying CA to free text would require bespoke loss functions' was intended to refer to the experiments, not the theory. By ``bespoke loss'' we meant only that, to instantiate Eq.(2) in a given application, one must specify a bounded guardrail loss $L(x,a)\in[0,1]$ (equivalently, a task-appropriate guardrail metric $g(x,a)$, e.g., violation rate or factuality error) for such an application. Once such a bounded loss is defined, CA selects $\hat\lambda$ and the finite-sample guarantee applies exactly as in the multiple choice setting.
>
> This also explains our Section 4 terminology of  actions and contexts: it was chosen to cover not only question answering but also agentic deployments, where an LLM (or LLM+tools) selects among environment actions. CA’s guarantees apply directly in this view: the Primary proposes a finite set $A(x)$, CA forms the $\lambda$-relaxed set $C_\lambda(x)$, and the Guardian (possibly a human) selects within that restricted slate under a risk budget $\alpha$. While suitable public benchmarks for fully agentic evaluations are limited, the formulation and guarantees are designed with such potential applications in mind. Our experiments  use multiple choice question-style benchmarks as controlled proxies. Notably, Section~5.2 (helpfulness--harmlessness) already goes beyond binary right/wrong by using discrete safety severities (e.g., 0--3), illustrating CA on non-binary correctness outcomes.
>
> Free text within CA: minimal recipe.
>
> For given calibration context $x_i$ we can have an arbitrarily large but finite potential output set $A(x_i)$. Sample $K$ free-text candidates from the Primary (e.g., across reasoning budgets/prompts/beam) and inherently consider these to be the $K$ top scoring $p(x_i,a)$. Call this $A_K \in A_i$. Record Primary/Guardian scores $\{p(x_i,a)\}$ for $\{a \in A_K\}$ and $\{g(x_i,a)\}$ for $\{a\in A_K\}$. Sample the Guardian model for its output and consider this to be $\arg\max_{a \in A(x_i)} g(x_i,a)$. Then we are able to define the bounded guardrail loss $L_i(\lambda)$ as in Eq. 2, compute $\widehat R_n(\lambda)=\tfrac1n\sum_i L_i(\lambda)$, and select the smallest $\hat\lambda$ satisfying the CRC inequality in Eq. 3. If no  $\hat\lambda$ satisfied the CRC inequality then we would need to increase $K$ and re-run.
>
> Deployment (per input; Sec. 4.3/Alg. 1). For a new $x$: (i) obtain $p(x,a)$ for all $a\in A(x)$; (ii) form $C_{\hat\lambda}(x)=\{a:\,p(x,a)\ge \max_{a'}p(x,a')-\hat\lambda\}$; (iii) if $|C_{\hat\lambda}(x)|=1$, return its unique element (commit to the Primary); otherwise (iv) query the Guardian only on $C_{\hat\lambda}(x)$ and output $\arg\max_{a\in C_{\hat\lambda}(x)} g(x,a)$. This mirrors Sec. 4 exactly and carries the same finite-sample guardrail guarantee from CRC.
>
> Note that this does not change any core mechanism behind how CA works and does not require any changes to it as stated, it is just a specific proposal for eliciting the scores $p(x,a)$ and $g(x,a)$ for the calibration set and applying the CA framework. If instead the scores were easily obtainable using a functional form or some other method then that would also suffice. The main point is that at its technical foundations Conformal Arbitrage does not pertain just to questions that are in multiple choice style settings, but can truly apply to any setting with contexts and finite action sets.
>
> On the ``specialized Guardian is always better'' concern.
>
> CA’s risk guarantee does not assume the Guardian uniformly dominates the Primary. The CRC guarantee is on the final output (after our policy either commits or escalates) and holds at finite sample under exchangeability. If the Guardian adds little value on a subset of inputs (including when the Guardian is a human), the calibrated $\hat\lambda$ becomes small and escalations are rare; the guardrail bound is still met. When a performance gap exists, CA selectively exploits it while filtering the candidate slate, which budgets oversight effort.
>
> Our intended deployment target is a closed-API, black-box setting in which the user cannot train or host additional models on $x$ (the Primary may be substantially more capable or proprietary). Many learning-to-defer and model-routing methods assume joint training, access to logits/gradients, or router models—assumptions that do not match this deployment constraint. CA differs along three axes emphasized in the paper: (i) post-hoc, black-box operation (no joint training, no weights/logits), (ii) distribution-free, finite-sample guardrail control via CRC on a bounded loss (rather than optimizing an expected risk without distribution-free guarantees), and (iii) set-based escalation (forwarding only $C_{\hat{\lambda}}(x)$) that preserves cost/latency in closed-API use. Reflecting this target, our evaluation compares to single-model baselines and a cost-matched random router (the natural naive policy when only API calls are permitted), and includes an unrestricted ablation (CA$^\star$) that passes the full set to the Guardian to isolate the impact of restricted conformal sets. At comparable cost, CA recovers most of the stronger model’s gains while honoring the guardrail budget, which is precisely the objective in the black-box setting we study.
>
> On Theorem 1 regularity
>
> Eq. (2) defines a per-episode loss $L_i(\lambda)$ that is stepwise in $\lambda$, so the empirical $\widehat R_n(\lambda)$ is piecewise constant. Theorem 1 is a population-level result about $R(\lambda)=\mathbb{E}[L_i(\lambda)]$.
>
> We do agree, however, that it may not hold in practice that this population-level loss curve is strictly decreasing. It is the case if the score-gap thresholds are atom-free (e.g., in practice by adding an independent continuous infinitesimal tie-break to the Primary scores), then $R$ is \emph{continuous}. Strict decrease, however, additionally requires that enlarging $C_\lambda(x)$ can improve the guardrail on a set of inputs of positive probability; without that, $R$ may have flat regions. In such cases, the CRC risk control is unchanged, and a variant of Theorem 1 holds.
>
> If $R$ is merely non-increasing and right-continuous, define the leftmost feasible point
> $\lambda_\star=\inf\{\lambda\in\Lambda:\,R(\lambda)\le \alpha\}$. The CRC selector $\hat\lambda$ (the smallest $\lambda$ with
> $\tfrac{n}{n+1}\widehat R_n(\lambda)+\tfrac{B}{n+1}\le \alpha$) satisfies $R(\hat\lambda)\le \alpha$ for all $n$ (finite-sample control), and under the law of large numbers for bounded $L_i(\lambda)$ one has $\widehat R_n(\lambda)\to R(\lambda)$ pointwise for each $\lambda$. Any limit point $\bar\lambda$ of $\hat\lambda$ must then satisfy $R(\bar\lambda)\le \alpha$; with the “leftmost feasible” tie-break used in our definition, $\hat\lambda\to \lambda_\star$ in probability. If, in addition, the utility $U$ is non-increasing and continuous, then $U(\lambda_\star)-U(\hat\lambda)\to 0$.
> Essentially, when $R$ is strictly decreasing with slope bounded away from $0$ near $\alpha$, Theorem 1 further gives the $O(n^{-1})$ rate via the inverse slope; without that margin, we can still state consistency without a rate.
>
> We can include this clarification and the variant statement in the appendix of the camera-ready. We thank the reviewer for their close reading and eliciting this clarification.
>
> Minor comments (acknowledged)
>
> We will correct the noted typos and notational slips: remove the duplicated 'the,' clarify that the conformal predictor attains coverage, write $\hat{\lambda}$-relaxed set, fix $g(x,a)$ in Algorithm 1, and change set of contexts to set of actions. We thank the reviewer for their close reading and identification of each of these typos.
>
> We hope these clarifications address the reviewer’s concerns and help recalibrate the overall assessment.

---

> > ### Comment · Reviewer_E6o9 · 2025-08-05
> > **Question about minimal recipe for free text within CA**
> >
> > To me, it seems that the main problem of utilizing your conformal arbitrage method beyond simple predefined output spaces (e.g., multi-choice questions, action sets in agentic settings) is that the output space (action set A(x) in the paper’s methodology) in free text task is very large to run the proposed algorithm.
> >
> > In this regard, the minimal recipe you propose for free text seems to be an approximation of the calibration step of your algorithm since it uses only K action/output samples $A_K(x)$ during calibration and not the full action set $A(x)$. During deployment, you write “obtain $p(x,a)$ for all $a\in A(x)$” which also seems infeasible for large action sets $A(x)$. I would be happy if you could clarify this, maybe I am missing something?

---

> > ### Comment · Reviewer_E6o9 · 2025-08-06
> > **Question regarding the guardian model**
> >
> > I understand the point the authors made in their rebuttal and I think they should also clarify this in the paper, my concern is slightly different. It is my understanding that conformal risk control only provides a marginal guarantee. Thus, if a guardian model (such as a human) has heterogenous performance across inputs, it might lead to the CA method performing worse. The experimental set up does not cover this case, so it is unclear how the performance might degrade.

---

> > ### Comment · Reviewer_E6o9 · 2025-08-06
> > **On Theorem 1 regularity**
> >
> > To be honest, I was not able to follow all the details in the clarification for Theorem 1, but I believe the authors that clarification is correct and addresses the issue regarding $R$ being continuous and strictly decreasing. I do think it is valuable that the authors include the clarification at least in the appendix as it not straightforward.

---

> ### Author Response · Authors · 2025-08-06
> **Response to "Question about minimal recipe for free text within CA"**
>
> Thank you for the great question. This is a nuanced point. To recall the main problem we are trying to solve, we want to achieve as much primary utility as possible while staying within a guardian's guardrail objective. For the problem of free-text generation, we can indeed achieve this with the "minimal recipe" approach provided in our rebuttal, even when the universe of all possible actions can be massive. Specifically, let us note that the calibration step can provides a verifiable guarantee that for any $K$ (even small ones) we achieve the desired guardian loss, or not. If such a guardian loss is achieved, we have achieved our goal.
>
> Specifically, for a fixed set size $K$ and a fixed generation policy, calibration verifies whether the CRC inequality in Eq.(3) is feasible.
> If it is, we already have a finite-sample guarantee that the final (commit/escalate) policy obeys the guardrail budget $\alpha$; in other words, we have achieved the guardian objective while achieving as much primary utility as possible and there is no need to score beyond the $K$ candidates the Primary proposed. If the inequality is not feasible, we simply increase $K$ and/or diversify the slate and re-calibrate.
>
> This is not an approximation of CA. The algorithm itself is unchanged; we are instantiating it on the finite action set induced by the Primary’s slate, with exchangeability preserved between calibration and deployment.
>
> However, we acknowledge that there could be hypothetical examples where to achieve the desired guardian objective, $K$ needs to be very large.This possibility is covered by our theory; it does not affect the validity of the guarantees. However, such examples are highly unrealistic and there is even evidence in the literature to support this claim. Empirically, recent work shows that even naive repeat sampling often finds high-quality answers with modest $K$ (tens to low hundreds; see, e.g., prior results on repeat sampling for LLMs [1]).
>
> Our slate construction is at least as strong as naive repeat sampling: if one chooses the generation policy to be “draw $K$ i.i.d. samples,” CA’s calibration and guarantees apply verbatim; if, instead, one asks the Primary to produce its top-$K$ or a diversified set across reasoning budgets/prompts, the resulting slate weakly dominates the naive pool for the same budget, and CA can only benefit from the higher-quality proposals.
>
> In short: calibration tells us, for the chosen $K$ and generation policy, whether the safety budget $\alpha$ is met. If it is, there is no reason to consider more of the (latent) action space—the Primary has already surfaced candidates it believes best, and CA ensures the guardrail objective on the final decision. If it is not, increase $K$ or diversify the slate and re-calibrate; the guarantees remain intact throughout.
>
> [1] Large Language Monkeys: Scaling Inference Compute with Repeated Sampling, Brown et al.
>
>
> Additionally, we'd like tothank the reviewer for explicitly noting that CA applies to agentic action sets. This is an important part of our contribution and motivation: we aim at safe deployment of agents in high-stakes decision-making scenarios where a language model (or LLM+tools) must act autonomously most of the time, yet escalate selectively. In this regime, CA serves as a principled, black-box controller: the Primary proposes actions, CA enforces a score-gap filter and a calibrated risk budget $\alpha$, and escalation to a human or stronger model occurs only when needed. This provides a rigorous mechanism for scalable oversight.

---

> ### Author Response · Authors · 2025-08-06
> **Technical details on how "Response to 'Question about minimal recipe for free text within CA'" is covered by CA as stated**
>
> Here is a more technical explanation of the above.
>
> Let $x \in \mathcal{X}$ and let $\mathcal{Y}_L$ be all strings up to a fixed length $L$.
> Under a fixed generation policy (top-$K$ by the Primary’s internal scoring, or $K$ proposals across budgets/prompts/beam), the Primary produces a finite slate $S(x)=\{a_1,\ldots,a_K\}\subseteq \mathcal{Y}_L$.
>
> We define the Primary score on $\mathcal{Y}_L$ by setting $p(x,a) =$ model-provided score for $a \in S(x)$, and $p(x,a) = -\infty$ for $a \not \in S(x)$. So off-slate strings are implicitly ruled out by the score-gap router.
>
> For each calibration context $x_i$:
> (i) build $S(x_i)$ with the same generation policy;
> (ii) obtain $p(x_i,a)$ for $a\in S(x_i)$;
> (iii) query the Guardian once to produce its own free-text response $y_G(x_i)$
>
> Obtain $g(x_i,a)$ for $a\in S(x)$ and for $y_G(x_i)$.
> Then compute the bounded guardrail loss exactly in the form of Eq.(2):
> $$
> L_i(\\lambda)\= g(x_i, y_G(x_i))-\\max_{a\in C_\lambda(x_i)} g(x_i,a)\ \in[0,B],
> $$
> the empirical risk $\widehat R_n(\lambda)=\tfrac{1}{n}\sum_{i=1}^n L_i(\lambda)$, and select the smallest $\hat\lambda$ satisfying Eq.(3).
>
> No scores over “all strings’’ are required—only the Primary’s scores on the slate and the Guardian’s scores on the finite $S(x)$ (with $y_G(x_i)$ providing the natural baseline).
>
> Deployment
>
> For a new $x$: form $S(x)$ with the same generation policy, compute $p(x,a)$ for $a\in S(x)$, construct $C_{\hat\lambda}(x)$, and (i) commit if $|C_{\hat\lambda}(x)|=1$, else (ii) query the Guardian only on $C_{\hat\lambda}(x)$ and output $\arg\max_{a\in C_{\hat\lambda}(x)} g(x,a)$.
> Because the generation policy is identical in calibration and deployment, the episodes are exchangeable and the CRC guarantee applies verbatim.
>
> Key takeaways.
>
> (1) Everything above is exactly the CA specification in Section 4; nothing in the theory or algorithm changes.
>
> (2) This recipe does not conflict with the original specification. If one did have scores for the entire action set, those could be used directly and CA would operate unchanged. The ``minimal recipe'' is simply a computationally efficient instantiation for free-text settings, not a modification of CA. All of the same guarantees are maintained.
>
> (3) If Eq.3 is feasible for a chosen $K$, the guardrail objective is already met; if not, increase $K$ and re-calibrate, guarantees remain intact throughout.
>
> We hope this clarifies how CA applies to free-text generation and appreciate the reviewer’s engagement with this point.

---

> > ### Comment · Reviewer_E6o9 · 2025-08-06
> >
> > Thank you for the clarification, this is much more understandable. I have two questions:
> > - For this part "$p(x,a) =$ model-provided score", do you mean that the LLM would score it's own output in the generated answer a? Or generate a first and score it separately?
> > - Eq. (2) in the paper is slightly different from the equation you show here: In the paper the maximum for $g(x_i,a)$ is taken over a in $A(x_i)$ and here it is only $g(x_i, y_G(x_i))$. What would be the effect of this change?
> >
> > Overall, to me this generalization to free text doesn't seem trivial. I understand that the theory is more general but these details that need to be taken into account to apply the algorithm beyond predefined action sets are not really present in the paper (e.g., sampling of S(x), change in Eq. 2). Clarifying these details for free text application and having experiments with this application would definitely strengthen the contributions of the paper. In my opinion, it is also fine to state that the paper only concentrates on applications with predefined action sets, but, like I wrote in the review, it would be nice to clarify this earlier on in the paper (e.g., the introduction) so the paper is more cohesive.

---

> > > ### Author Response · Authors · 2025-08-07
> > >
> > > Thank you for the continued discussion, we truly appreciate your engagement and push to dig deep into our work.
> > >
> > > As to the first question, yes the LLM scores its own outputs as in the Section 5 experiments.
> > >
> > > We'd like to clarify that the equation we show above doesn't not actually differ from Equation 2; rather, we have that $g(x_i, y_G(x_i)) := \max_{a \in A_i(x)} g(x_i, a)$ since $y_G(x_i)$ is the output determined by the Guardian itself which is principally designed to maximize the guardrail metric.
> > >
> > > We appreciate the concern that extending CA to free-text may not appear straightforward. From the perspective of our theory, however, free-text is an instantiation rather than a generalization. We emphasize that our paper covers free text generation from the theoretical perspective, and our above discussion shows how to instantiate an approach to implementing CA that saves on compute, but does not require any generalization to the equations or theory as originally stated. We will clarify in the revision how free text generation is included in our current theoretical results. And then in the appendix discuss how to instantiate it, under low compute settings.
> > >
> > > We also emphasize that free-text generation—though a common LLM application—is only one of many settings our methodology addresses. More broadly, CA applies to decision-making problems where an agent must choose among actions, providing a principled, black-box mechanism for scalable oversight with finite-sample guardrail guarantees.
> > >
> > > That being said, we completely understand the interest and relevance in free-text generation applications. We appreciate the reviewer's direct suggestions in this direction. We will do the following: as recommended by the reviewer, we will provide a discussion early on in the paper indicating that the main intended use cases is decision making under defined action sets, but that bespoke instantiations are possible for free-text generation problems as well. We will then point to a section in the appendix where we explain the full details in a crystallized presentation of our above minimal recipe and discussion. We hope that the reviewer would agree that this addresses their concerns with the cohesion of the paper.
> > >
> > > Again we truly appreciate the engagement by the reviewer and hope that they find our responses satisfactory enough to consider a more positive assessment of our paper. Thank you again.

---

> > > > ### Comment · Reviewer_E6o9 · 2025-08-07
> > > >
> > > > I appreciate that the author will clarify their focus is decision making under defined action sets and add details for a possible for free-text application in the appendix.
> > > >
> > > > One more clarification, for this part:
> > > >
> > > > "since $y_G(x_i)$ is the output determined by the Guardian itself which is principally designed to maximize the guardrail metric"
> > > >
> > > > This is not necessarily always true? If the Guardian model is an LLM as well then $y_G(x_i)$ would just be a sample free-text output from the LLM and not necessarily maximize the guardrail metric (i.e., "(iii) query the Guardian once to produce its own free-text response $y_G(x_i)$"), otherwise one would have to maximize over the whole free-text output space of the Guardian LLM. That's why I was wondering why you can change Eq. 2 to only take into account a single sample and not the maximum.

---

> > > > > ### Author Response · Authors · 2025-08-08
> > > > >
> > > > > We are happy to hear that the reviewer appreciates our clarification of focus ond decision making and our addition of an appendix section on free-text generation. We appreciate the reviewer’s positive impact in these improvements to the paper.
> > > > >
> > > > > Thank you for the followup clarification question. First we need to recall that the scores are generated directly by the mmodel itself in its output (not from logits etc). Within this instantiation of CA to free-text generation, we  thus are able to define $g(x,y_{G}(x))$ to be the maximum score, i.e. the, perhaps temperature 0 for consistency, output of the LLM powering the guardian model itself is defined to have the highest guardrail score for this context. This could be enforced in a number of ways when obtaining scores from the same Guardian model on other outputs $a$, for example by adding to the prompt that the model already assigned highest possible score to $y_{G}(x)$ or by capping any score to be no higher than $g(x,y_{G}(x))$.
> > > > >
> > > > > In particular, we do not need to “maximize over the whole free-text output space of the Guardian LLM”. Within this specified instantiation of CA to free-text generation all of our methodology works as stated and guarantees hold as stated by taking $max_{a} g(x,a)$ to be $g(x,y_{G}(x))$. Thus we are not changing Eq. 2 at all; we are just choosing a g within this instantiation for which we know that $g(x,y_{G}(x))$ achieves the maximum needed for Eq. 2.
> > > > >
> > > > > Similar to our above discussion, we emphasize that our paper covers free text generation from the theoretical perspective, and that similar to our discussion of obtaining primary scores, this is a particular instantiation and design choice that falls within the general CA framework and allows us to handle free-text generation. In other words, it is a modeling choice (particular instantiation) for trying to capture whatever the user’s true intent is in terms of the guardrail metric; repeat sampling at a higher temperature to obtain $y_{G}(x)$ or some other more sophisticated approach are also valid modeling choices and should be seen as different instantiations of CA to this free-text generation problem. We are proposing one particular instantiation and showing that it falls within the CA framework while mitigating the computational concerns.
> > > > >
> > > > > We hope this provides further clarification to the reviewer, and we will be sure to incorporate such concerns into our proposed appendix section on free-text generation. We sincerely thank the reviewer for their constructive comments, which have helped us improve the paper’s presentation. We hope that our revisions and detailed responses will merit a more favorable evaluation of our work.

---

> > > > > > ### Comment · Reviewer_E6o9 · 2025-08-08
> > > > > >
> > > > > > I understand that the simple recipe is an instantiation of CA for free text outputs and other instantiations are possible, but I am trying to understand if this simple recipe would work/makes sense as the authors would add it to the paper/appendix. For example, saying $g(x,y_{G}(x))$ is defined to be the maximum score doesn't make a lot of sense to me (or at least seems like a strong assumption). Just because $y_{G}(x)$ is the most likely output of the Guardian model doesn't mean it is the best output in terms of the guardrail objective. Sure, one can make the model artificially score $y_{G}(x)$ the highest but then the distribution of these scores is not the same as the distribution of the scores of the Guardian model without constraints which should be the one that actually best aligns with the guardrail objective.

---

> > > > > > > ### Author Response · Authors · 2025-08-08
> > > > > > >
> > > > > > > We thank the reviewer for the continued engagement, and for the opportunity to further discuss our work.
> > > > > > >
> > > > > > > We would like to clarify that in our framework we would not consider, for example, GPT 4.1 to be the "Guardian model," but rather the underlying LLM (in this case 4.1) and whatever else goes into computing the required scores. For example, both the Primary and the Guardian could be GPT 4.1, but with different system prompts, or with different transformations applied to the score output by the model. Again recall that is not that we are accessing logits or some other characteristic of the underlying LLM, but rather in our experiment section directly ask the model for its raw score. But if you consider Section 5.2 on PKU-SafeRLHF you can see an example of not using an LLM for the Guardian.
> > > > > > >
> > > > > > > Recall that then such a Guardian model assigns the guardrail score itself, it does not come from an outside source. So it is completely valid for it to assign itself a 1 (max score) to whatever output it gives on a given prompt for temperature 0.
> > > > > > >
> > > > > > > That being said, we completely agree that ``just because $y_{G}(x)$ is the most likely output of the Guardian model doesn't mean it is the best output in terms of the guardrail objective.'' If we take the guardrail objective to be the more intuitive interpretation of what, we a human user, deploying a Guardian model to achieve scalable oversight or routing, wants to achieve. We 100% agree with this. We were just taking this to be one possible instantiation. Repeat sampling would likely lead to better alignment to the goal of the human user.
> > > > > > >
> > > > > > > We hope this further clarifies both CA as a whole and in particular its applications to free-text generation. We look forward to the opportunity to lay out all these details on free-text generation in an appendix section.

---

> > > > > > > > ### Author Response · Authors · 2025-08-08
> > > > > > > > **empirical proof of concept for free-text generation**
> > > > > > > >
> > > > > > > > Given interest across reviewers of seeing result for free-text generation, we have worked hard to obtain a proof of concept of such empirical results. We still maintain that the paper's main strengths lie in the results that were originally presented, though we would be happy to include an expanded scope of these empirical results in the appendix section that we promised to add in our response to Reviewer E6o9.
> > > > > > > >
> > > > > > > > We use **OpenAI's HealthBench**. There are 5 axes on which the responses to the prompts in this benchmark are scored: communication quality, instruction following, accuracy, context awareness, and completeness. There is a natural tension between communication quality, which measures the plain language understandability of the response, and accuracy, which measures clinical factuality. We treat these as the ``competing objectives'' for our Conformal Arbitrage framework.
> > > > > > > >
> > > > > > > >
> > > > > > > >
> > > > > > > > We instantiate the Conformal Arbitrage (CA) framework on HealthBench by treating communication quality as the primary objective and accuracy as the guardian objective. For each prompt $x$ in the benchmark, we generate using GPT 4.1 mini a slate $S(x)$ of $K=10$ candidate responses, $5$ prompted to focus on  communication quality and $5$ prompted to focus on accuracy.
> > > > > > > >
> > > > > > > > The Primary model assigns each candidate $a \in S(x)$ a predicted communication-quality score $p(x,a)$, while the Guardian model assigns each candidate $a \in S(x)$ a predicted communication-quality score $g(x,a)$. Both are prompted like ``predicted probability this output would get full marks for communication (accuracy)." Given a relaxation parameter $\lambda \ge 0$, we define the $\lambda$-relaxed conformal set. $\lambda$ is calibrated on a held-out calibration set with loss Following Eq.(2), the calibration loss for a context $x$ and relaxation $\lambda$ is $L(\lambda) = g\big(x,y_G(x)\big) - \max_{a \in C_{\lambda}(x)} g(x,a)$, where $y_G(x)$ is the Guardian model's own preferred output (its argmax over all possible outputs by $g$ as discussed with Reviewer E6o9).
> > > > > > > >
> > > > > > > > For $\alpha = 0.15$, CA achieves a communication quality of $0.699$ and accuracy of $0.640$. The communication-oracle baseline attains the highest communication ($0.884$) but similar accuracy ($0.634$), while the accuracy-oracle baseline maximizes accuracy ($0.778$) at lower communication quality ($0.659$).
> > > > > > > >
> > > > > > > > **Disclaimer.** We have not had the opportunity to thoroughly validate these results or run ablations.
> > > > > > > > They are based on a limited sample size (only $100$ prompts each for calibration and testing)
> > > > > > > > and are intended solely as a **proof of concept** for applying CA in this free-text generation setting.
> > > > > > > > We do not view extensive experiments in such domains as being within the scope of this first paper;
> > > > > > > > however, we will gladly expand upon these empirical results in a subsection of the appendix on free-text generation.
> > > > > > > >
> > > > > > > > We appreciate the reviewers’ engagement and feedback. Our aim in the main text was to present CA as a general, black-box controller with finite-sample guarantees, illustrated through two representative axes (cost–accuracy and helpfulness–harmlessness). Given CA’s breadth, a full sweep across every possible domain isn’t practical for this first paper, but we believe the Section 5 studies capture the core strengths of the approach.
> > > > > > > >
> > > > > > > > That said, we’re very pleased with the constructive discussion with Reviewer E6o9, and we will include the new free-text proof-of-concept and clarifications in an appendix of the camera-ready. We hope that both the discussion above and this new **proof of concept empirical demonstration of CA on free-text generation will help reviewers view our contribution more favorably.**

---

> > > > > > > > ### Comment · Reviewer_E6o9 · 2025-08-08
> > > > > > > >
> > > > > > > > Thanks for the clarification, I did not realize that the prompt is part of the Primary/Guardian model (this might also be worth clarifying in the paper). Then, setting a specific output as highest score makes more sense, but yes sampling should give you better alignment with the guardrail objective.
> > > > > > > >
> > > > > > > > I like the direction of the experiments for the free-text applications, it would be nice to see if with more samples the authors can improve the performance of CA, since the initial proof of concept numbers show only a marginal improvement.
> > > > > > > > That being said, I am not sure we can take the proof of concept into account in our evaluation since it is after the rebuttal deadline. Maybe the AC could clarify this?
> > > > > > > >
> > > > > > > > I think the free-text experiments will further strengthen the paper, but I am already willing to increase the score (without taking the free-text experiments into account).

---

> > > > > > > > > ### Author Response · Authors · 2025-08-08
> > > > > > > > >
> > > > > > > > > We are happy to hear that we were able to clarify that point with respect to the Primary and Guardian models and that it helped to understand our point about the specific output as the highest score; we will certainly clarify this in the paper as you suggest.
> > > > > > > > >
> > > > > > > > > We will be sure to include experiments expanding on this proof of concept for the free-text application into the aforementioned appendix section, with increased sample size etc.
> > > > > > > > >
> > > > > > > > > We're very glad to hear that you’re willing to increase your score, even independent of the free-text experiments. If you feel sufficiently comfortable, we’d be sincerely grateful if you could update your official review to reflect this.
> > > > > > > > >
> > > > > > > > > We’ve really appreciated the chance to discuss our work in this level of detail during the rebuttal; your questions have genuinely helped us clarify and improve the paper.

---

### Official Review · Reviewer_2XWC · 2025-07-02

**Clarity:** 3
**Significance:** 3
**Originality:** 3
**Rating:** 4
**Confidence:** 4

**Summary:**

The paper is focusing on the problem of tensions between potentially competing objectives in LLM models (for example, helpfulness versus harmlessness, cost versus accuracy, and reward versus safety). The authors propose Conformal Arbitrage, a post-hoc API framework based on conformal risk control that learns a data-driven threshold to mediate between a model optimized for a primary objective and a more conservative guardian (another model or a human domain expert). The authors test their framework in two trade-off settings, i.e., cost-accuracy axis and helpfulness-harmlessness axis on TruthfulQA, PKU-SafeRLHFand MMLU datasets. Comparisons against single-model baselines and random routers show that the proposed framework outperforms cost- and  risk-matched random routing, recovers most gains of the stronger model at a fraction of the cost, and works with closed-API deployments without accessing weights or logits.

**Questions:**

How would you design a task specific loss function for applying your framework to free-form text generation?

**Ethical Concerns:**

["NO or VERY MINOR ethics concerns only"]

**Final Justification:**

Thanks to the authors for their answer. I will keep my score.

**Limitations:**

The authors acknowledge the limitations of their work and important directions to further explore for extending their framework to varied tasks.

**Quality:**

3

**Strengths And Weaknesses:**

Strengths:

The proposed algorithm is logical, elegant and theoretically grounded - instead of rejecting an uncertain instance, escalate it to a more conservative model. The Conformal Arbitrage framework  works with black box APIs, requires no access to logits or probabilities, and only makes one call to the primary model, and when the call is routed just one call to the guardian model.

The authors include ablation analysis which shows that the Conformal Arbitrage frontier is stable overall. Results demonstrate that the framework traces an efficient frontier between helpfulness and harmlessness.

Extensive review of related work, the framework is well placed in the context of the literature.

The paper is well written and argumented, and addresses an important open problem in the literature.


Weaknesses:

The framework can only be applied to multiple choice tasks and does not work with free-form text generation, which is a limitation of the proposed approach.

The authors do not explain why they focus only on single-step, two-model router  (Primary and Guardian); while they acknowledge that deeper cascades are possible, they do not explore this in the paper - it is likely this could yield improved results or finer control.

While Conformal Arbitrage is a lightweight and theoretically grounded approach, its current form is best suited for structured tasks and simple model pairs. The authors leave enhancements in generalization to complex tasks and richer model configurations for future work.

---

> ### Author Rebuttal · Authors · 2025-07-31
>
> We sincerely thank the reviewer for the thoughtful and positive assessment.
>
> Scope beyond multiple choice.
>
> Conformal Arbitrage (CA) is not restricted to multiple choice. The theory in Section 4 is stated for any input-dependent finite action slate $A(x)$; Algorithm 1 and CRC calibration (Eq. (3)) operate on such action sets regardless of how candidates are produced (MCQ options or free-text proposals).
>
> Our limitations sentence 'our study is confined to multiple-choice tasks; applying CA to free text would require bespoke loss functions' was intended to refer to the experiments, not the theory. By ``bespoke loss'' we meant only that, to instantiate Eq.(2) in a given application, one must specify a bounded guardrail loss $L(x,a)\in[0,1]$ (equivalently, a task-appropriate guardrail metric $g(x,a)$, e.g., violation rate or factuality error) for such an application. Once such a bounded loss is defined, CA selects $\hat\lambda$ and the finite-sample guarantee applies exactly as in the multiple choice setting.
>
> This also explains our Section 4 terminology of  actions and contexts: it was chosen to cover not only question answering but also agentic deployments, where an LLM (or LLM+tools) selects among environment actions. CA’s guarantees apply directly in this view: the Primary proposes a finite set $A(x)$, CA forms the $\lambda$-relaxed set $C_\lambda(x)$, and the Guardian (possibly a human) selects within that restricted slate under a risk budget $\alpha$. While suitable public benchmarks for fully agentic evaluations are limited, the formulation and guarantees are designed with such potential applications in mind. Our experiments  use multiple choice question-style benchmarks as controlled proxies. Notably, Section~5.2 (helpfulness--harmlessness) already goes beyond binary right/wrong by using discrete safety severities (e.g., 0--3), illustrating CA on non-binary correctness outcomes.
>
> Free text within CA: minimal recipe.
>
> For given calibration context $x_i$ we can have an arbitrarily large but finite potential output set $A(x_i)$. Sample $K$ free-text candidates from the Primary (e.g., across reasoning budgets/prompts/beam) and inherently consider these to be the $K$ top scoring $p(x_i,a)$. Call this $A_K \in A_i$. Record Primary/Guardian scores $\{p(x_i,a)\}$ for $\{a \in A_K\}$ and $\{g(x_i,a)\}$ for $\{a\in A_K\}$. Sample the Guardian model for its output and consider this to be $\arg\max_{a \in A(x_i)} g(x_i,a)$. Then we are able to define the bounded guardrail loss $L_i(\lambda)$ as in Eq. 2, compute $\widehat R_n(\lambda)=\tfrac1n\sum_i L_i(\lambda)$, and select the smallest $\hat\lambda$ satisfying the CRC inequality in Eq. 3. If no  $\hat\lambda$ satisfied the CRC inequality then we would need to increase $K$ and re-run.
>
> Deployment (per input; Sec. 4.3/Alg. 1). For a new $x$: (i) obtain $p(x,a)$ for all $a\in A(x)$; (ii) form $C_{\hat\lambda}(x)=\{a:\,p(x,a)\ge \max_{a'}p(x,a')-\hat\lambda\}$; (iii) if $|C_{\hat\lambda}(x)|=1$, return its unique element (commit to the Primary); otherwise (iv) query the Guardian only on $C_{\hat\lambda}(x)$ and output $\arg\max_{a\in C_{\hat\lambda}(x)} g(x,a)$. This mirrors Sec. 4 exactly and carries the same finite-sample guardrail guarantee from CRC.
>
> Note that this does not change any core mechanism behind how CA works and does not require any changes to it as stated, it is just a specific proposal for eliciting the scores $p(x,a)$ and $g(x,a)$ for the calibration set and applying the CA framework. If instead the scores were easily obtainable using a functional form or some other method then that would also suffice. The main point is that at its technical foundations Conformal Arbitrage does not pertain just to questions that are in multiple choice style settings, but can truly apply to any setting with contexts and finite action sets.
>
> Why a two-model, single-step router:
>
> Although CA could support deeper commit/escalate cascades, we focus on a two-model, single-step policy to keep the mechanism transparent. CA does, in principle, admit multi-stage cascades in which a single calibrated $\hat{\lambda}$ still provides finite-sample control of the final guardrail loss; one can also use stage-specific thresholds $\{\lambda_t\}$ calibrated jointly use multi-lambda conformal risk control. While cascades are a natural fit for cost–accuracy trade-offs, for helpfulness–harmlessness the notion of “depth’’ is less canonical, and the restricted-conformal set mechanism is the primary lever we wished to isolate. Our core objective here is to balance two competing metrics under closed-API constraints, rather than to identify the cheapest model capable of answering a query; moreover, the Guardian-as-human interpretation corresponds to a one-shot escalation with an explicit risk budget $\alpha$. For these reasons, we adopt the two-model setting as a principled and sufficient testbed for our theory and black-box deployment goals, noting that the methodology extends to deeper cascades without changes to the underlying analysis.
>
> We hope that the above responses adequately address the reviewer's questions, especially regarding the scope beyond multiple choice, and that they further inform your already favorable review. We thank the reviewer again for their time and engagement with our work.

---

### Official Review · Reviewer_ypLq · 2025-07-05

**Clarity:** 3
**Significance:** 2
**Originality:** 3
**Rating:** 4
**Confidence:** 2

**Summary:**

This paper presents a data-driven approach called Conformal Arbitrage (CA), rooted in conformal risk control theory. This method allows for precise control over trade-offs, enabling users to explicitly specify how much they are willing to compromise on one objective in order to gain on another. Notably, CA operates entirely at the API level, without the need for access to model logits or adjustments to model weights. Experiments conducted with the GPT-4.1 series demonstrate Pareto improvements of CA over random model selection strategies.

**Questions:**

1. In Algorithm 1, how can we choose an appropriate lambda value based on different data distributions and tasks?
2. Regarding the first cost versus accuracy task, can similar results be achieved for long-to-short tasks? For instance, in "Towards Thinking-Optimal Scaling of Test-Time Compute for LLM Reasoning" (2025). For the second task, instead of assigning a score to a response candidate, can the method be extended to safety generation tasks?
3. In Figure 1, when alpha = 0.25, could you explain why the performance of CA is lower than that of random routing?

**Ethical Concerns:**

["NO or VERY MINOR ethics concerns only"]

**Limitations:**

Please see my questions.

**Quality:**

3

**Strengths And Weaknesses:**

1. CA functions as a lightweight router that mediates between a primary model optimized for a specific objective and a more conservative Guardian model.
2. The promising results indicate that the proposed method effectively leverages the performance gap between specialized models, yielding superior outcomes compared to naive model selection.

---

> ### Author Rebuttal · Authors · 2025-07-31
>
> We thank the reviewer for highlighting that Conformal Arbitrage (CA) effectively leverages performance gaps between models via a lightweight router, and for noting that this yields improvements over naive selection. We emphasize that CA extends beyond “model selection”: the Guardian may be a human, enabling scalable oversight with explicit risk budgeting.
>
> Viewing the Guardian as a human links CA to scalable oversight and control in AI safety. With a calibration set, primary and guardrail metrics, Primary outputs, and human assessments, CA provides finite-sample guarantees that the safety (guardrail) metric, measured against human judgments, stays within budget while allowing the Primary to act autonomously whenever possible. Section 5.2 (PKU SafeRLHF) illustrates the helpfulness–harmlessness trade-off. More broadly, the human-Guardian perspective is a theoretical contribution to AI safety beyond model selection/routing.
>
> Q1. Choosing $\hat{\lambda}$ in Algorithm 1.
>
> Conformal Arbitrage uses conformal risk control to select $\hat{\lambda}$ directly from data. Given a calibration set and the bounded guardrail loss in Eq.(2), we apply Eq. (3) to choose the smallest $\lambda$ whose adjusted empirical risk is at most the target $\alpha$. This yields finite-sample, distribution-free control of the guardrail.
>
> Q2. Long-to-short (test-time compute) and safety generation.
>
> CA extends naturally to variable test-time compute: the Primary proposes multiple candidates under different reasoning budgets (short/medium/long). Take the primary metric to be compute (tokens/latency/cost) to be minimized and the guardrail to be task performance (e.g., accuracy) with a bounded loss. CA forms the $\hat{\lambda}$-relaxed set $C_{\hat{\lambda}}(x)$ from Primary scores $p(x,a)$ (any logits-free quality/confidence proxy; one may also use a monotone transform favoring lower-compute candidates) and uses CRC to calibrate the smallest $\hat{\lambda}$ achieving the target accuracy level on the final output. For safety generation, replace “accuracy” with a bounded safety loss; CA then uses CRC to calibrate $\hat{\lambda}$ to satisfy the safety budget while letting the Primary act autonomously when appropriate.
>
> Q3. Why is CA below random at $\alpha=0.25$ in Fig.1?
>
> Appendix B.5 (Fig. 3, Table 6) provides insight on this question. At $\alpha=0.25$, calibration yields $\hat{\lambda}$ of about $0.277$ and a mean conformal set size of about $1.457$, so $C_{\hat{\lambda}}(x)$ is often a singleton. Under our restricted-conformal set passing policy, the Guardian receives only $C_{\hat{\lambda}}(x)$; if the correct answer was pruned, it cannot be recovered. In contrast, the random router would provide the full set of potential answers, yielding a slight advantage at this single budget. The unrestricted ablation (CA$^\star$) confirms the mechanism: when the Guardian sees the full set of potential answers, CA outperforms random at comparable cost. We adopt the restricted-set policy by design to bias the Guardian toward high-primary-utility options under a guardrail budget.
>
> We hope these clarifications are helpful and address the reviewer’s questions.

---

> > ### Comment · Reviewer_ypLq · 2025-08-06
> >
> > Thank you for the detailed explanation. Regarding the application to generation tasks, empirical results would help confirm the generalizability of CA. For now, I maintain my initial scores.

---

### Note · Authors · 2025-08-13

We thank the reviewers for their positive and supportive reviews. In particular, we thank Reviewer E6o9 for the engaging discussion, which helped us clarify and highlight the strengths and generality of Conformal Arbitrage. We hope the other reviewers and the AC will read this exchange and recognize that Conformal Arbitrage is not restricted to multiple-choice questions, as some initially inferred. Given that the two other reviewers already initially provided positive assessments despite this misunderstanding, we hope this clarification will further positively inform both their and the AC’s final evaluations.

---

### Decision · Program_Chairs · 2025-09-17

**Decision:**

Accept (poster)

**Comment:**

Conformal Arbitrage (CA) proposes a black-box way to balance utility and guardrails: it calibrates a CRC threshold to decide when to commit to a Primary model versus escalate to a more conservative Guardian (model or human), yielding finite-sample control of a bounded guardrail metric while preserving much of the Primary’s utility. The setup is deployment-relevant (API-only, no logits/weights), and the experiments trace an efficient frontier and beat cost-matched random routing; the ablation also explains the lone budget point where random does better (the restricted conformal set often collapses to a singleton). The main limitations are scope and baselines: evaluation leans on multiple-choice–style proxies; free-text and richer agentic settings are argued for but not fully demonstrated; and stronger black-box router comparisons would sharpen the case. Presentation previously blurred scope (now clarified), and open questions remain about heterogeneous Guardian quality and deeper cascades. During rebuttal, the authors clarified that the theory applies to any finite action slate, gave a minimal free-text instantiation, addressed regularity details, and promised edits; reviewers converged on borderline-accept after these clarifications.

It is expected that the authors will make the necessary changes in the final version to incorporate the main takeaways from the author-reviewer discussion, including  (1) early, explicit statement of scope (finite action slates) and how free-text is instantiated; (2) add the Theorem-1 regularity clarification; and (3) if feasible, add a stronger black-box router baseline and a brief analysis of Guardian heterogeneity sensitivity.